# GRAM: GENERALIZATION IN DEEP RL WITH A ROBUST ADAPTATION MODULE

## ABSTRACT

The reliable deployment of deep reinforcement learning in real-world settings requires the ability to generalize across a variety of conditions, including both in-distribution scenarios seen during training as well as novel out-of-distribution scenarios. In this work, we present a framework for dynamics generalization in deep reinforcement learning that unifies these two distinct types of generalization within a single architecture. We introduce a robust adaptation module that provides a mechanism for identifying and reacting to both in-distribution and out-of-distribution environment dynamics, along with a joint training pipeline that combines the goals of in-distribution adaptation and out-of-distribution robustness. Our algorithm GRAM achieves strong generalization performance across in-distribution and out-of-distribution scenarios upon deployment, which we demonstrate on a variety of realistic simulated locomotion tasks with a quadruped robot.

## 1 INTRODUCTION

Due to the diverse and uncertain nature of real-world settings, generalization is an important capability for the reliable deployment of data-driven, learning-based frameworks such as deep reinforcement learning (RL). Policies trained with deep RL must be capable of generalizing to a variety of different environment dynamics at deployment time, including both familiar training conditions and novel unseen scenarios, as the complex nature of real-world environments makes it difficult to capture all possible variations in the training process.

Existing approaches to zero-shot dynamics generalization in deep RL have focused on two complementary concepts: adaptation and robustness. Contextual RL techniques (Hallak et al., 2015) learn to identify and adapt to the current environment dynamics to achieve the best performance, but this adaptation is only reliable for the range of in-distribution (ID) scenarios seen during training. Robust RL methods (Nilim & Ghaoui, 2005; Iyengar, 2005), on the other hand, maximize the worst-case performance across a range of possible environment dynamics, providing generalization to out-of-distribution (OOD) scenarios at the cost of conservative performance in ID environments.

This work shows how to extract the benefits of these complementary approaches in a unified framework called GRAM: Generalization in deep RL with a Robust Adaptation Module. GRAM achieves generalization to both ID and OOD environment dynamics at deployment time within a single architecture. Our main contributions are as follows:

1. We introduce a *robust adaptation module* in Section 4 that provides a mechanism for identifying both ID and OOD environment dynamics within the same architecture. We extend existing contextual RL approaches by using an epistemic neural network (Osband et al., 2023) to incorporate a measure of uncertainty about the environment at deployment time.

2. We propose a joint training pipeline in Section 5 that combines a teacher-student architecture for learning adaptive ID performance with adversarial RL training for robust OOD performance, resulting in a single unified policy that can achieve both ID and OOD dynamics generalization.

3. We demonstrate the strong ID and OOD performance of GRAM in Section 7 through comprehensive experiments on realistic simulated locomotion tasks with the Unitree Go2 quadruped robot in Isaac Lab (Mittal et al., 2023).

## 2 RELATED WORK

Zero-shot generalization in deep RL has received significant attention in recent years (see Kirk et al. 2023 for a survey). The most common formulation of this problem is contextual RL (Hallak et al., 2015), and many studies have demonstrated the importance of leveraging contextual information in deep RL to improve generalization (e.g., Benjamins et al. 2023). For the case of known contexts, both model-free (Beukman et al., 2023) and model-based (Prasanna et al., 2024) architectures have been proposed to effectively incorporate context into training. When the context is unknown at deployment time, it must be inferred from past observations. Some domain randomization methods directly consider a policy conditioned on history (Peng et al., 2018; Tiboni et al., 2024), while self-supervised approaches infer context from history through the use of variational inference (Yang et al., 2020; Chen et al., 2022; Ren et al., 2022), dynamics prediction (Lee et al., 2020b), or separation-based objectives (Luo et al., 2022). Similar techniques have also been applied in the related area of meta RL (Nagabandi et al., 2019; Rakelly et al., 2019; Zintgraf et al., 2020), which require additional learning in the deployment environment for adaptation. Supervised approaches, on the other hand, leverage privileged context information during training, and apply a teacher-student architecture to train a policy that can be deployed using only the history of past observations (Lee et al., 2020a; Kumar et al., 2021; Margolis et al., 2024). All of these methods are designed to adapt across the range of ID contexts seen during training, but are not specifically trained to handle OOD contexts with different environment dynamics. As a result, their ability to generalize is sensitive to the distribution of ID training contexts, and they may not generalize well to OOD contexts.

Robust RL focuses on generalizing to OOD environments at deployment time by maximizing worst-case performance over a set of transition models (Nilim & Ghaoui, 2005; Iyengar, 2005). Deep RL methods most commonly apply robustness during training through the use of parametric uncertainty or adversarial training. Parametric uncertainty methods consider a range of simulation parameters during training, and optimize for worst-case performance across these environments (Rajeswaran et al., 2017; Mankowitz et al., 2020). Adversarial RL instead applies worst-case perturbations during training to provide robustness to unknown dynamics or disturbances at deployment time. Various types of adversarial interventions have been considered, including external forces (Pinto et al. 2017; Reddi et al. 2024; Xiao et al. 2024) as well as perturbations to actions (Tessler et al., 2019), observations (Zhang et al., 2020), and transitions (Queeney & Benosman, 2023; Queeney et al., 2024). Robust RL methods have also been applied to capture estimation uncertainty (Xie et al., 2022) and context shifts (Lin et al., 2020; Ajay et al., 2022; Zhang et al., 2023) in contextual RL and meta RL. These approaches provide robust generalization to environment dynamics that were not explicitly seen during training, but often sacrifice ID performance to achieve robustness.

In this work, we are interested in both ID and OOD generalization. The possibility of different modes at deployment is related to methods that train a collection of policies to select from at deployment time. Many learning-based approaches to safety switch between a task-based policy and a recovery policy in order to guarantee safety at deployment time (Thananjeyan et al., 2021; Wagener et al., 2021; Contreras et al., 2024; He et al., 2024; Sinha et al., 2024). Other methods select between a finite collection of different behaviors based on the deployment environment (Ajay et al., 2022; Chen et al., 2023). Contextual RL can also be viewed as learning a collection of policies for ID adaptation, and our robust adaptation module extends this approach to incorporate a mode for OOD generalization as well.

## 3 PROBLEM FORMULATION

**Contextual RL** We model the problem of dynamics generalization in deep RL as a Contextual Markov Decision Process (CMDP) (Hallak et al., 2015). A CMDP considers a set of contexts $\mathcal{C}$ that define a collection of MDPs $\{\mathcal{M}_c\}_{c \in \mathcal{C}}$. For each $c \in \mathcal{C}$, we have an MDP given by the tuple $\mathcal{M}_c = (\mathcal{S}, \mathcal{A}, p_c, r, \rho_0, \gamma)$, where $\mathcal{S}$ is the set of states, $\mathcal{A}$ is the set of actions, $p_c : \mathcal{S} \times \mathcal{A} \times \mathcal{C} \to P(\mathcal{S})$ is the context-dependent transition model where $P(\mathcal{S})$ represents the space of probability measures over $\mathcal{S}$, $r : \mathcal{S} \times \mathcal{A} \to \mathbb{R}$ is the reward function, $\rho_0$ is the initial state distribution, and $\gamma$ is the discount rate. We focus on the setting where the transition model $p_c$ depends on the context $c \in \mathcal{C}$ (i.e., varying dynamics), while the reward function $r$ remains the same across contexts (i.e., same task). For a policy $\pi$ and context $c \in \mathcal{C}$, performance is given by the expected total discounted

returns

$$J(\pi, c) = \mathbb{E}_{\tau \sim (\pi, c)} \left[ \sum_{t=0}^{\infty} \gamma^t r(s_t, a_t) \right], \tag{1}$$

where $\tau = (s_0, a_0, s_1, \ldots)$ and $\tau \sim (\pi, c)$ represents a trajectory sampled by deploying the policy $\pi$ in the MDP $\mathcal{M}_c$.

**Problem statement**  We assume that the context is available as privileged information during training, but is not available for deployment. This is often the case when a policy is trained in simulation and deployed in the real world. In addition, due to unknown factors at deployment time, it is typically not possible to train across all possible contexts. Instead, we assume access to a subset of ID training contexts $\mathcal{C}_{\text{ID}} \subset \mathcal{C}$, and we write $c \sim \mathcal{C}_{\text{ID}}$ to represent a sample from a training distribution over ID contexts. We define $\mathcal{C}_{\text{OOD}} = \mathcal{C} \setminus \mathcal{C}_{\text{ID}}$ as the set of OOD contexts that are not seen during training. *Our goal is to train a single policy that performs well in both ID and OOD contexts at deployment time*:

$$\max_{\pi} J(\pi, c) \quad \forall c \in \mathcal{C} = \mathcal{C}_{\text{ID}} \cup \mathcal{C}_{\text{OOD}}. \tag{2}$$

Because we have access to ID contexts during training, we can directly maximize performance for $c \in \mathcal{C}_{\text{ID}}$. On the other hand, we do not have access to OOD contexts during training, so we instead seek to achieve robust generalization for $c \in \mathcal{C}_{\text{OOD}}$ by applying techniques from robust RL. Note that adaptive ID performance and robust OOD performance represent two distinct types of generalization with different objectives, making it challenging to achieve both with a single policy.

## 4  ROBUST ADAPTATION MODULE

We build upon the teacher-student architecture for adaptation in deep RL, which has demonstrated strong performance in complex robotics applications (Lee et al., 2020a; Kumar et al., 2021; Margolis et al., 2024). However, because this approach focuses on adaptation across ID contexts observed during training, its OOD generalization capabilities depend strongly on the relationship between $\mathcal{C}_{\text{ID}}$ and $\mathcal{C}_{\text{OOD}}$. For OOD contexts with environment dynamics that are not similar to ID training contexts, we show in our experiments that the standard teacher-student architecture can perform poorly. In order to generalize to both ID and OOD contexts at deployment time, we introduce a *robust adaptation module* that explicitly incorporates a mechanism for identifying and reacting to OOD contexts.

**Teacher-student training**  The teacher-student approach to generalization in deep RL assumes access to $c_t \in \mathcal{C}_{\text{ID}}$ at every timestep throughout training, and leverages this privileged context information to train a teacher policy that can adapt to different contexts. The teacher policy applies a context encoder $f : \mathcal{C} \to \mathcal{Z}$ that maps the context to a latent feature $z_t = f(c_t)$, which is then provided as an input to the policy $\pi : \mathcal{S} \times \mathcal{Z} \to P(\mathcal{A})$ and critic $V^\pi : \mathcal{S} \times \mathcal{Z} \to \mathbb{R}$. In this work, we consider the latent feature space $\mathcal{Z} = \mathbb{R}^d$. The context encoding $z_t = f(c_t)$, policy $\pi(a_t \mid s_t, z_t)$, and critic $V^\pi(s_t, z_t)$ are trained to minimize the average actor-critic RL loss over ID contexts given by

$$\mathcal{L}_{\text{RL}} = \mathbb{E}_{c \sim \mathcal{C}_{\text{ID}}} \left[ \mathcal{L}_\pi(c) + \mathcal{L}_V(c) \right], \tag{3}$$

where $\mathcal{L}_\pi(c)$ and $\mathcal{L}_V(c)$ represent the policy loss and critic loss, respectively, of a given RL algorithm for the context $c \sim \mathcal{C}_{\text{ID}}$. In our experiments, we apply Proximal Policy Optimization (PPO) (Schulman et al., 2017) as the RL algorithm. See the Appendix for details.

Note that the teacher policy cannot be applied at deployment time because it requires privileged information about the context $c_t$ in order to compute the latent feature $z_t$. For this reason, RL training is followed by a supervised learning phase where a student policy is trained to imitate the teacher policy using only the recent history of states and actions from the last $H$ timesteps $h_t = (s_{t-H}, a_{t-H}, \ldots, s_t) \in \mathcal{H}$. In particular, an adaptation module $\phi : \mathcal{H} \to \mathcal{Z}$ that maps recent history to a latent feature $\hat{z}_t = \phi(h_t)$ is trained to minimize the loss

$$\mathcal{L}_{\text{enc}} = \mathbb{E}_{c \sim \mathcal{C}_{\text{ID}}} \left[ \mathbb{E}_{\tau \sim (\pi, c)} \left[ \|f(c_t) - \phi(h_t)\|^2 \right] \right] = \mathbb{E}_{c \sim \mathcal{C}_{\text{ID}}} \left[ \mathbb{E}_{\tau \sim (\pi, c)} \left[ \|z_t - \hat{z}_t\|^2 \right] \right], \tag{4}$$

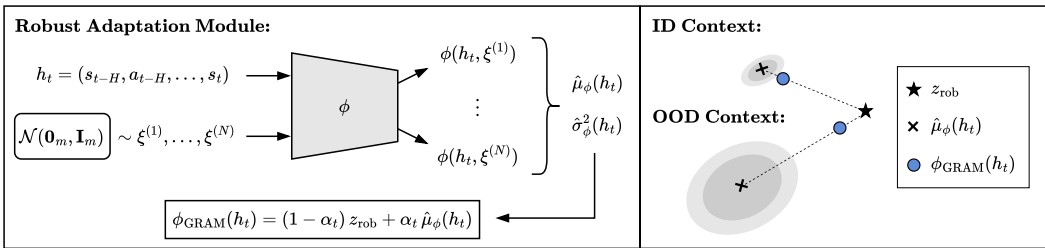

Figure 1: Robust adaptation module used by GRAM at deployment time. Left: Epistemic neural network $\phi$ outputs a sample mean and variance of latent feature estimates for a history $h_t$, which are used to calculate $\phi_{\text{GRAM}}$ in (8). Right: In ID contexts, variance of latent feature estimates will be low and $\phi_{\text{GRAM}}$ will be close to the mean estimate. In OOD contexts with different environment dynamics, variance will be high and $\phi_{\text{GRAM}}$ will output an estimate close to $z_{\text{rob}}$.

where expectation is taken with respect to trajectories sampled using the student policy in ID contexts $c \sim \mathcal{C}_{\text{ID}}$ during training. This training represents a form of implicit system identification across ID contexts. Using the history encoding $\hat{z}_t = \phi(h_t)$, the policy $\pi(a_t \mid s_t, \hat{z}_t)$ can be applied at deployment time because it does not require privileged information as input.

**Robust adaptation module** In the teacher-student architecture, the adaptation module $\phi$ is trained to identify ID contexts from history, resulting in strong performance across $c \in \mathcal{C}_{\text{ID}}$. However, because it is only possible to train on ID contexts, the adaptation module $\phi$ and resulting policy $\pi(a_t \mid s_t, \hat{z}_t)$ may not generalize well to OOD contexts $c \in \mathcal{C}_{\text{OOD}}$ that were not seen during training. Because $\phi$ is a learned module, its output $\hat{z}_t$ is only reliable for the distribution of history inputs $h_t$ that were observed during training. This suggests that $\hat{z}_t = \phi(h_t)$ may not be useful if the environment dynamics of OOD contexts result in trajectories that are different from the ID trajectories seen during training.

In order to quantify the level of uncertainty present in the latent feature estimate $\hat{z}_t$, in this work we represent the adaptation network $\phi$ as an epistemic neural network (Osband et al., 2023) with the form

$$\phi(h_t, \xi) = \phi_{\text{base}}(h_t) + \phi_{\text{epi}}(\tilde{h}_t, \xi), \tag{5}$$

where $\xi \in \mathbb{R}^m$ is a random input that is sampled from a multivariate standard Gaussian distribution $\xi \sim \mathcal{N}(\mathbf{0}_m, \mathbf{I}_m)$, and $\tilde{h}_t$ represents an intermediate feature of $h_t$ constructed from $\phi_{\text{base}}$ with gradients stopped. See the Appendix for details. By incorporating a random input in the second component of (5) (i.e., the "epinet"), this architecture provides a distribution of latent feature estimates for a history input $h_t$ rather than a single point estimate. See the left side of Figure 1 for an illustration. For a history $h_t$ and $N$ random input samples $\xi^{(1)}, \ldots, \xi^{(N)} \sim \mathcal{N}(\mathbf{0}_m, \mathbf{I}_m)$, we can write the sample mean and variance of the latent feature estimates as

$$\hat{\mu}_\phi(h_t) = \frac{1}{N} \sum_{i=1}^{N} \phi(h_t, \xi^{(i)}), \quad \hat{\sigma}_\phi^2(h_t) = \frac{1}{N-1} \sum_{i=1}^{N} (\phi(h_t, \xi^{(i)}) - \hat{\mu}_\phi(h_t))^2, \tag{6}$$

where all operations are performed per-dimension. We train (5) to minimize the encoder loss across random input samples, resulting in the modified encoder loss given by

$$\mathcal{L}_{\text{enc}}^{\text{GRAM}} = \mathbb{E}_{\xi \sim \mathcal{N}(\mathbf{0}_m, \mathbf{I}_m)} \left[ \mathbb{E}_{c \sim \mathcal{C}_{\text{ID}}} \left[ \mathbb{E}_{\tau \sim (\pi, c)} \left[ \|f(c_t) - \phi(h_t, \xi)\|^2 \right] \right] \right]. \tag{7}$$

By doing so, the variance of the latent feature estimates will be low over the distribution of history inputs $h_t$ that were seen during training (i.e., trajectories sampled from ID contexts), but not for histories in OOD contexts with different dynamics that were not seen during training.

Using the epistemic neural network architecture in (5), we introduce a *robust adaptation module* to generalize to both ID and OOD contexts at deployment time. When uncertainty of the latent feature estimates is low, we output the mean estimate $\hat{\mu}_\phi(h_t)$ to allow for adaptation in ID contexts. When uncertainty of the latent feature estimates is high, we bias the mean estimate towards a special robust

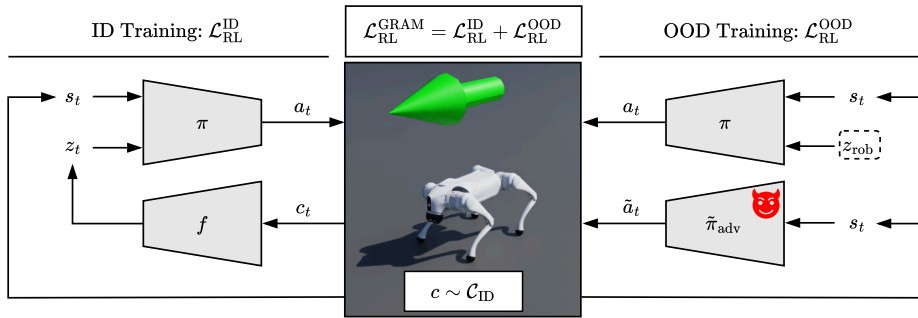

Figure 2: Joint RL training pipeline used by GRAM, which combines standard ID data collection and adversarial OOD data collection for every RL update. Training environments are assigned to *ID training* or *OOD training* at each iteration, and assignments alternate between iterations.

latent feature $z_{\text{rob}}$ to identify that OOD dynamics have been detected. For a given history $h_t$, our robust adaptation module $\phi_{\text{GRAM}} : \mathcal{H} \to \mathcal{Z}$ outputs a latent feature $\hat{z}_t = \phi_{\text{GRAM}}(h_t)$ according to

$$\phi_{\text{GRAM}}(h_t) = (1 - \alpha_t) \, z_{\text{rob}} + \alpha_t \, \hat{\mu}_\phi(h_t), \quad \alpha_t = \exp\left(-\beta(\|\hat{\sigma}_\phi(h_t)\|^2 - c)_+\right), \tag{8}$$

where $(\,\cdot\,)_+ = \max(\,\cdot\,, 0)$ and $\alpha_t \in [0, 1]$, with $\alpha_t \to 1$ when uncertainty is low and $\alpha_t \to 0$ when uncertainty is high. We include a scale parameter $\beta$ and shift parameter $c$ to allow for easy finetuning of $\alpha_t$ based on the output magnitude of the trained epinet in ID contexts, which can be calculated using a validation set of $\|\hat{\sigma}_\phi(h_t)\|^2$ values collected at the end of training. See the Appendix for details.

The robust latent feature $z_{\text{rob}}$ defines an anchor point in latent feature space where we can apply robust training. Because we consider a single unified policy architecture, note that the RL loss $\mathcal{L}_{\text{RL}}$ in (3) incentivizes the privileged context encoding $z_t = f(c_t)$ to move away from $z_{\text{rob}}$ if it will lead to better performance. Given this intuition, we propose the use of $z_{\text{rob}} = \mathbf{0}_d$ in our implementation of GRAM to allow the outputs of a randomly initialized context encoder to start near $z_{\text{rob}}$ at the beginning of training and move away from it as needed. Then, at deployment time we bias the latent feature estimate back towards the robust anchor point $z_{\text{rob}}$ if the estimate is unreliable due to OOD environment dynamics. See the right side of Figure 1 for an illustration.

By using a special robust latent feature $z_{\text{rob}}$, we incorporate the failure mode of OOD dynamics directly into the existing adaptation framework as a single instance in latent feature space. This allows us to leverage the existing training procedure and architecture for adaptation in ID contexts, while applying tools from robust RL to encode robust behavior into $\pi(a_t \mid s_t, z_{\text{rob}})$ for OOD generalization *within the same architecture*. The use of a single architecture for implicit regularization benefits of this robust training also allows for implicit regularization beyond $z_{\text{rob}}$. In the next section, we provide details on the robust RL techniques that we use during training to achieve OOD generalization.

## 5 Training for Robust Adaptation

Our robust adaptation module provides an intuitive structure for achieving both ID and OOD dynamics generalization within a single architecture. In order to accomplish this goal, we jointly train our policy $\pi(a_t \mid s_t, z_t)$ for adaptive performance in ID environments and robust performance in OOD environments (i.e., when $z_t = z_{\text{rob}}$). For a given iteration of RL training, we assign each training environment to either *ID training* or *OOD training*. This assignment determines how the latent feature vector is calculated, as well as how data collection occurs in the environment. See Figure 2 for an overview.

**Data collection** Within each iteration, all data for a given training environment is collected according to either standard ID data collection or adversarial OOD data collection, as described in the following paragraphs. This provides temporal consistency when training the policy for adaptive or robust performance, respectively. However, we alternate these assignments between iterations,

which allows full trajectories to contain a mixture of both forms of data. As we show in our experiments, this mixed data collection design provides additional robustness benefits compared to using the same training assignment for entire trajectories.

**Training for ID adaptation**  For environments assigned to *ID training*, we follow the same data collection and training updates as the standard teacher-student architecture. We calculate the privileged latent feature vector as $z_t = f(c_t)$, and perform standard data collection with $\pi(a_t \mid s_t, z_t)$. We update the policy $\pi$, critic $V^\pi$, and context encoder $f$ according to (3), and we denote this loss by $\mathcal{L}_{\text{RL}}^{\text{ID}}$.

**Training for OOD robustness**  For environments assigned to *OOD training*, we use the robust latent feature vector $z_{\text{rob}}$ and apply an adversarial RL training pipeline to provide robustness to worst-case environment dynamics at deployment time. We perform data collection with $\pi(a_t \mid s_t, z_{\text{rob}})$, and we introduce an adversary policy $\tilde{\pi}_{\text{adv}} : \mathcal{S} \rightarrow P(\tilde{\mathcal{A}}_{\text{adv}})$ that is trained with RL to minimize the returns of $\pi(a_t \mid s_t, z_{\text{rob}})$. Note that the adversary policy $\tilde{\pi}_{\text{adv}}(\tilde{a}_t \mid s_t)$ does not need to share the same action space as our policy (i.e., $\tilde{a}_t \in \tilde{\mathcal{A}}_{\text{adv}} \neq \mathcal{A}$). In our simulation experiments on the Unitree Go2 quadruped robot, the adversary applies external forces to the robot's body a percentage of the time during data collection, where the direction of the external force is learned by the adversary. Using this adversarially collected data, we update the policy $\pi$ and critic $V^\pi$ according to (3) with the robust latent feature vector $z_{\text{rob}}$ as input. We denote this loss by $\mathcal{L}_{\text{RL}}^{\text{OOD}}$. Note that adversarial RL provides a method for sampling worst-case trajectories during training, which leads to robust generalization in unseen OOD contexts at deployment time when the robust latent feature $z_{\text{rob}}$ is provided to the policy as input.

**Training the robust adaptation module**  As in the standard teacher-student architecture, RL training is followed by a supervised learning phase where we train the adaptation module for deployment using data collected with the student policy. We accomplish this by training the epistemic neural network architecture in (5) on the modified encoder loss $\mathcal{L}_{\text{enc}}^{\text{GRAM}}$ in (7). We apply the same ID and OOD training assignments used in RL training, where ID data collection uses the student policy $\pi(a_t \mid s_t, \hat{z}_t)$ with $\hat{z}_t = \hat{\mu}_\phi(h_t)$ and OOD data collection uses $\pi(a_t \mid s_t, z_{\text{rob}})$. We do not apply an adversary during the supervised learning phase, as the goal is to train the epinet in (5) to output estimates with low variance in ID contexts.

## 6 GRAM ALGORITHM

Together, the robust adaptation module in (8) and the training procedure in Section 5 form our algorithm GRAM. GRAM combines standard ID data collection and adversarial OOD data collection during training to optimize the RL loss

$$\mathcal{L}_{\text{RL}}^{\text{GRAM}} = \mathcal{L}_{\text{RL}}^{\text{ID}} + \mathcal{L}_{\text{RL}}^{\text{OOD}}, \tag{9}$$

followed by a supervised learning phase to optimize the encoder loss $\mathcal{L}_{\text{enc}}^{\text{GRAM}}$. Finally, by applying the robust adaptation module $\phi_{\text{GRAM}}$ at deployment time, our policy achieves both ID and OOD dynamics generalization within a single unified architecture.

There are several key factors that allow GRAM to achieve strong performance in both ID and OOD environments. First, the special robust latent feature $z_{\text{rob}}$ allows us to separate the competing goals of ID adaptation and OOD robustness during training, while still considering a unified policy structure. Second, the robust adaptation module provides a mechanism for identifying and reacting to unreliable latent feature estimates in OOD scenarios at deployment time. Finally, we see in our experiments that the joint training pipeline with mixed data collection provides additional robustness benefits to the policy.

As we show in the next section, we found that the use of a teacher-student architecture for ID adaptation and adversarial RL for OOD robustness result in strong performance on the simulated quadruped robot locomotion tasks we consider in our experiments. However, note that it is also possible to apply GRAM with different choices of contextual RL methods for ID adaptation and robust RL methods for OOD generalization, which represents an interesting avenue for future work.

Table 1: Base ID context set

| Parameter | Dimension | Nominal | Range |
|---|---|---|---|
| Friction multiple | 1 | 1.00 | $[0.25, 2.00]$ |
| Added base mass (kg) | 1 | 0.00 | $[-1.00, 3.00]$ |
| Motor strength multiple | 12 | 1.00 | $[0.80, 1.20]$ |
| Joint angle bias (rad) | 12 | 0.00 | $[-0.10, 0.10]$ |

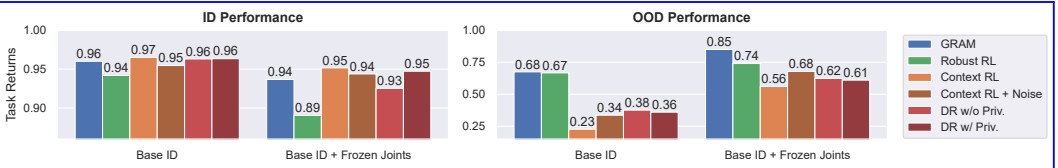

Figure 3: Average normalized task returns in ID and OOD contexts. Performance averaged across contexts shown in Figure 4.

## 7 EXPERIMENTS

We evaluate the performance of GRAM on realistic simulated locomotion tasks with the Unitree Go2 quadruped robot in Isaac Lab (Mittal et al., 2023). The goal of the robot is to track a velocity command $\mathbf{v}_t^{\mathrm{cmd}} = [v_x^{\mathrm{cmd}}, v_y^{\mathrm{cmd}}, \omega_z^{\mathrm{cmd}}] \in \mathbb{R}^3$ provided as input, where $v_x^{\mathrm{cmd}}, v_y^{\mathrm{cmd}}$ represent target forward and lateral linear velocities, respectively, and $\omega_z^{\mathrm{cmd}}$ represents a target yaw angular velocity. For each episode, we uniformly sample a forward linear velocity command between $0.5$ and $1.0$ meters per second (i.e., $v_x^{\mathrm{cmd}} \sim \mathcal{U}([0.5, 1.0]), v_y^{\mathrm{cmd}} = 0$) and calculate the yaw angular velocity command $\omega_z^{\mathrm{cmd}}$ throughout the episode based on a target heading direction. The simulated quadruped robot can be seen in Figure 2, with the velocity command represented by a green arrow.

The policy has access to noisy proprioceptive observations available from standard onboard sensors at every timestep (joint angles, joint velocities, projected gravity, and base angular velocities), and outputs target joint angles $a_t \in \mathbb{R}^{12}$ for each of the robot's 12 degrees of freedom that are converted to torques by a PD controller with proportional gain $K_p = 25$ and derivative gain $K_d = 0.5$. The maximum episode length is 20 seconds with target joint angles processed at 50 Hz, which corresponds to 1,000 timesteps per episode. We consider the reward function used in Margolis et al. (2024), which includes rewards for tracking the velocity command $\mathbf{v}_t^{\mathrm{cmd}}$ and regularization terms to promote smooth and stable gaits. See the Appendix for additional details.

We conduct experiments to investigate several hypotheses related to our algorithm GRAM:

**H1**. Existing contextual RL and robust RL methods demonstrate trade-offs between ID and OOD performance that depend critically on the set of contexts $\mathcal{C}_{\mathrm{ID}}$ seen during training.

**H2**. GRAM can achieve strong ID and OOD generalization with a single unified policy.

**H3**. GRAM identifies ID contexts from OOD contexts at deployment time in a way that automatically adjusts for different choices of $\mathcal{C}_{\mathrm{ID}}$.

**H4**. The unified architecture and joint training pipeline of GRAM outperforms other implementation choices for achieving ID and OOD generalization.

In order to test these hypotheses, we compare GRAM to contextual RL and robust RL on two different choices of training sets $\mathcal{C}_{\mathrm{ID}}$: (i) *Base ID* and (ii) *Base ID + Frozen Joints*. First, we consider the *Base ID* training set described in Table 1. This set represents moderate variations that are commonly used to promote sim-to-real transfer, and do not require significant adaptation to achieve good performance across ID contexts. Next, we consider the same variations shown in Table 1 while also freezing one (or none) of the robot's 12 joints. We refer to this set as *Base ID + Frozen Joints*, which represents a more diverse set of ID contexts with varying dynamics.

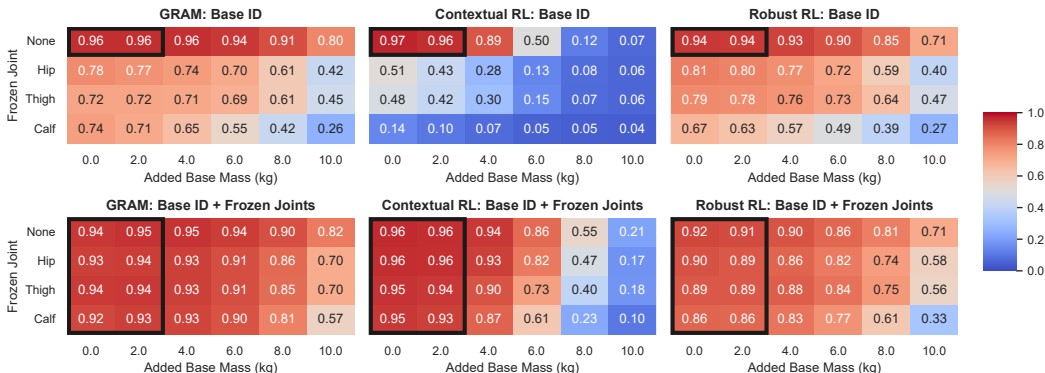

Figure 4: Average normalized task returns across a range of added base masses and frozen joint types at deployment time. Black boxes represent ID training contexts contained in $\mathcal{C}_{\text{ID}}$. Top: Training with *Base ID* context set. Bottom: Training with *Base ID + Frozen Joints* context set. For both choices of $\mathcal{C}_{\text{ID}}$, GRAM achieves ID performance comparable to contextual RL and OOD performance comparable to robust RL. Contextual RL fails to generalize well outside of $\mathcal{C}_{\text{ID}}$, while robust RL leads to conservative ID performance as the size of $\mathcal{C}_{\text{ID}}$ increases (bottom vs. top).

We summarize the ID and OOD performance of all algorithms in Figure 3. Every method is trained using PPO (Schulman et al., 2017) as the base RL algorithm with the parallel training scheme from Rudin et al. (2021). Contextual RL uses the standard teacher-student approach, and robust RL applies adversarial training using the same adversary as GRAM. We also include additional baselines in Figure 3. We consider contextual RL with random noise added to latent features $z_t$ during training which is related to smoothing methods for robustness (Kumar et al., 2022), as well as domain randomization with and without the use of privileged context information in the critic during training. See the Appendix for detailed results on these additional baselines. We include all implementation details in the Appendix, including network architectures and the values of all hyperparameters.[1]

**Performance in ID contexts**   The left side of Figure 3 shows ID performance for all algorithms, averaged across the ID contexts in Figure 4. The performance of GRAM, contextual RL, and robust RL across a range of frozen joint types and added base masses is shown in Figure 4, with ID contexts in black boxes. For both choices of training sets $\mathcal{C}_{\text{ID}}$, we see that contextual RL achieves the best ID performance and robust RL is the most conservative. All algorithms achieve strong ID performance for the *Base ID* training set, which does not require significant adaptation across contexts. As we move to a more diverse training set in the *Base ID + Frozen Joints* setting, however, the benefit of adaptation in contextual RL becomes more clear. In contrast, the non-adaptive robust RL method becomes overly conservative, providing evidence that supports our first hypothesis (**H1**). Conservative ID performance is the main drawback of robust RL methods, which can be meaningful in practical real-world settings where we expect to encounter ID conditions the majority of the time. Importantly, GRAM does not encounter the same ID performance issues as robust RL. Instead, GRAM is able to achieve ID performance comparable to contextual RL in both the *Base ID* and *Base ID + Frozen Joints* training settings.

**Performance in OOD contexts**   The right side of Figure 3 shows average OOD performance across the scenarios included in Figure 4 that were not seen during training. In contrast to the ID setting, contextual RL performs poorly in these OOD scenarios while robust RL achieves better generalization. In particular, we find that the OOD generalization capabilities of contextual RL strongly depend on the set of ID contexts seen during training. When trained on a more diverse range of contexts in the *Base ID + Frozen Joints* setting, the learned adaptation module in contextual RL is capable of generalizing to near-OOD conditions but fails in far-OOD scenarios. When trained on a narrow range of contexts in the *Base ID* setting, on the other hand, the performance of contextual RL rapidly declines outside of the set of ID contexts. This further validates our first hypothesis (**H1**). GRAM demonstrates strong OOD generalization in both training settings, outperforming all other

---

[1]Code is available at `https://anonymous.4open.science/r/iclr2025-gram`.

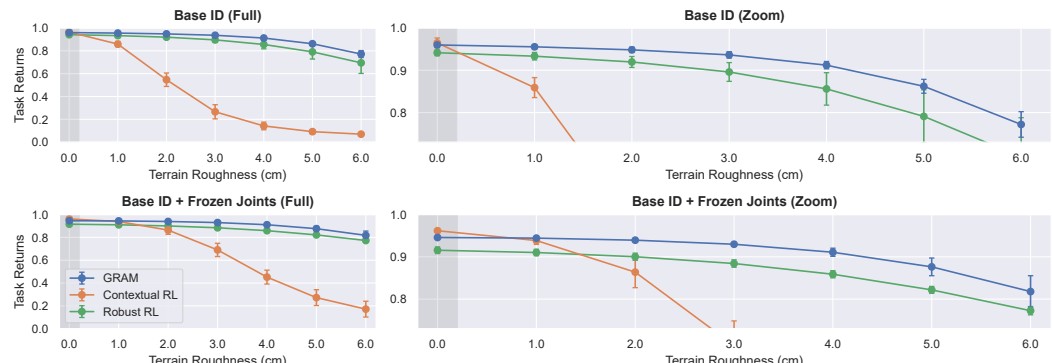

Figure 5: Average normalized task returns across a range of terrain roughness at deployment time. Training only occurs on flat terrain (0 cm roughness), shaded in grey. Error bars denote standard deviation across 5 training seeds. Top: Training with *Base ID* context set. Bottom: Training with *Base ID + Frozen Joints* context set. GRAM outperforms contextual RL and robust RL across terrains not seen during training.

methods including robust RL. GRAM leverages its adaptation capabilities to maintain strong performance in near-OOD contexts, while achieving robust performance similar to robust RL in far-OOD scenarios.

We also consider a separate OOD scenario in Figure 5 where the terrain roughness is varied at deployment time, compared to the flat terrain seen during training. In both of our training settings, contextual RL fails as terrain roughness increases. Robust RL and GRAM, on the other hand, only experience modest performance declines. We see that GRAM performs the best in rough terrain, achieving robust locomotion without being overly conservative. Together with GRAM's strong performance in ID contexts, this validates our second hypothesis (**H2**) that GRAM can generalize to ID and OOD settings using a single unified policy.

**Analysis of GRAM** Next, we perform additional analysis to better understand how our algorithm GRAM is able to achieve strong ID and OOD generalization. GRAM automatically adjusts to both ID and OOD contexts at deployment time through its robust adaptation module $\phi_{\text{GRAM}}$ in (8), which incorporates a measure of uncertainty about the deployment environment through the coefficient $\alpha_t \in [0, 1]$ ($\alpha_t \to 1$ for low uncertainty and $\alpha_t \to 0$ for high uncertainty). We plot the average GRAM coefficient $\alpha_t$ in Figure 6 for every deployment environment considered in our experiments, where we see that GRAM automatically adjusts to both the deployment environment and the set of ID training contexts. This provides evidence to support our third hypothesis (**H3**). GRAM outputs low uncertainty ($\alpha_t \to 1$) for ID contexts seen during training, and decreases $\alpha_t$ as OOD scenarios become increasingly novel compared to the training set. This is evident in the *Base ID* setting, where GRAM outputs high uncertainty (small $\alpha_t$) in OOD scenarios because it was only exposed to a limited set of contexts during training. When trained on a diverse range of contexts in the *Base ID + Frozen Joints* setting, on the other hand, GRAM automatically adjusts to this training set by applying larger values of $\alpha_t$ that correspond to less uncertainty.

Finally, we conduct an ablation study in Figure 7 to analyze the impact of GRAM's unified architecture and joint training pipeline. We compare GRAM to a variant that applies separate ID and OOD data collection to train a unified policy (instead of alternating training assignments between updates), as well as a modular approach that combines separate robust and adaptive policies at deployment time using an uncertainty threshold for switching. We see that these approaches can lead to comparable or slightly improved ID performance by isolating ID and OOD training, but this comes at the cost of robustness in OOD scenarios. GRAM's single unified policy and joint training pipeline provide additional robustness benefits that are difficult to achieve with more modular approaches, providing support for our fourth and final hypothesis (**H4**).

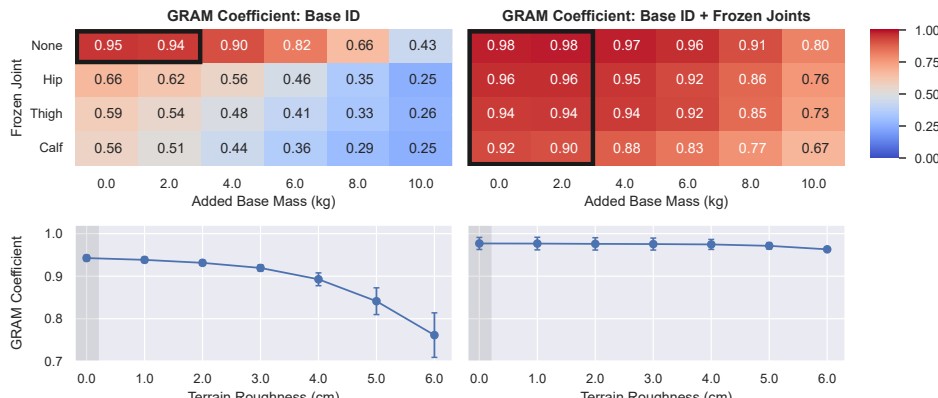

Figure 6: Average coefficient $\alpha_t$ used by GRAM at deployment time across a range of added base masses (top), frozen joint types (top), and terrain roughness (bottom). Error bars denote standard deviation across 5 training seeds. Left: Training with *Base ID* context set. Right: Training with *Base ID + Frozen Joints* context set. GRAM applies a smaller $\alpha_t$ on average in OOD contexts, automatically adopting more robust behavior in unfamiliar environments. When trained on a larger range of ID contexts (right vs. left), $\alpha_t$ increases on average because OOD contexts become more similar to ID contexts seen during training.

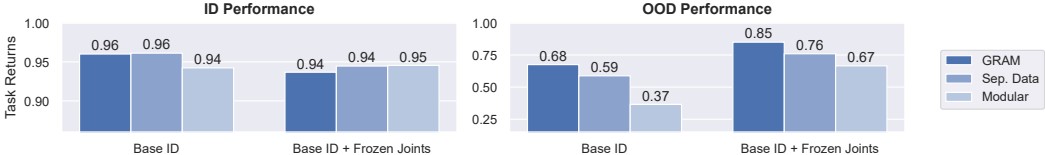

Figure 7: Ablation study for GRAM. Bars represent average normalized task returns in ID and OOD contexts. Performance averaged across contexts shown in Figure 4.

# 8 CONCLUSION

In this work, we have presented a deep RL framework that achieves both ID and OOD dynamics generalization at deployment time within a single architecture. Our algorithm GRAM leverages a robust adaptation module that allows for adaptation in ID contexts, while also identifying OOD environments with a special robust latent feature $z_{\mathrm{rob}}$. We presented a training pipeline that jointly trains for adaptive ID performance and robust OOD performance, resulting in strong generalization capabilities across a range of realistic simulated locomotion tasks on the Unitree Go2 quadruped robot in Isaac Lab. The ability to achieve ID and OOD generalization within a unified framework is critical for the reliable deployment of deep RL in real-world settings, and GRAM represents an important step towards this goal.

**Limitations and future work** Because OOD contexts are unknown during training by definition, the OOD generalization of GRAM depends on how well the robust RL training pipeline captures worst-case OOD dynamics. We applied a single choice of adversary in our experiments that worked well in practice, but it would be interesting to extend GRAM to incorporate different levels of robustness at deployment time. There are also opportunities to apply GRAM with different choices of contextual RL and robust RL techniques, extend our robust adaptation module to incorporate other methods for uncertainty quantification (Gawlikowski et al., 2023), and address other forms of generalization beyond dynamics related to modalities such as vision. Finally, we focused on simulated quadruped locomotion experiments in this work to conduct a comprehensive empirical study of GRAM across a variety of settings. We plan to deploy our GRAM framework in real-world hardware experiments on the Unitree Go2 quadruped robot as part of future work. We are also interested in applying GRAM to other applications where generalization is important, such as contact-rich manipulation tasks and agile quadrotor flight.

REPRODUCIBILITY STATEMENT

To ensure reproducibility, we provide access to the code used for our experiments at `https://anonymous.4open.science/r/iclr2025-gram`. We also provide implementation details related to our experimental setup and training procedure in Section 7 and Appendix A.

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

Table 2: Policy inputs

| Term | Notation | Dimension | Units | Noise |
|------|----------|-----------|-------|-------|
| **Observations** | | | | |
| Joint angle | $q_t$ | 12 | rad | $[-0.01, 0.01]$ |
| Joint velocity | $\dot{q}_t$ | 12 | rad / s | $[-1.50, 1.50]$ |
| Projected gravity | $g_t^{\mathrm{ori}}$ | 3 | - | $[-0.05, 0.05]$ |
| Base angular velocity | $\omega_t$ | 3 | rad / s | $[-0.20, 0.20]$ |
| Previous action | $a_{t-1}$ | 12 | rad | - |
| **Command** | | | | |
| Linear xy velocity command | $v_{xy}^{\mathrm{cmd}}$ | 2 | m / s | - |
| Angular yaw velocity command | $\omega_z^{\mathrm{cmd}}$ | 1 | rad / s | - |
| **Adaptation module** | | | | |
| Latent feature vector | $\hat{z}_t$ | 8 | - | - |

Table 3: Reward function from Margolis et al. (2024)

| Term | Equation | Weight |
|------|----------|--------|
| **Task** | | |
| Linear xy velocity | $\exp(-\|v_{xy} - v_{xy}^{\mathrm{cmd}}\|^2/0.25)$ | $1.0dt$ |
| Angular yaw velocity | $\exp(-(\omega_z - \omega_z^{\mathrm{cmd}})^2/0.25)$ | $0.5dt$ |
| **Stability** | | |
| Linear z velocity | $v_z^2$ | $-2.0dt$ |
| Angular roll-pitch velocity | $\|\omega_{xy}\|^2$ | $-0.05dt$ |
| Target height | $(h - 0.34)^2$ | $-30dt$ |
| Orientation | $\|g_{xy}^{\mathrm{ori}}\|^2$ | $-0.1dt$ |
| Undesired contacts | $\mathbf{1}_{\text{self-collision}}$ | $-1.0dt$ |
| Joint limits | $\|q - \mathrm{clip}(q, q_{\min}, q_{\max})\|_1$ | $-10dt$ |
| **Smoothness** | | |
| Joint torques | $\|\tau\|^2$ | $-1\mathrm{e}{-5}dt$ |
| Joint accelerations | $\|\ddot{q}\|^2$ | $-2.5\mathrm{e}{-7}dt$ |
| Action rate | $\|a_t - a_{t-1}\|^2$ | $-0.01dt$ |
| Feet air time | $\sum_{j=1}^{4}(t_{\mathrm{air},j} - 0.5) \cdot \mathbf{1}_{\text{new-contact},j}$ | $1.0dt$ |

Luisa Zintgraf, Kyriacos Shiarlis, Maximilian Igl, Sebastian Schulze, Yarin Gal, Katja Hofmann, and Shimon Whiteson. VariBAD: A very good method for Bayes-adaptive deep RL via meta-learning. In *Eighth International Conference on Learning Representations*, 2020.

## A    IMPLEMENTATION DETAILS

**Task definition**    For the simulated quadruped robot locomotion tasks we consider in our experiments, we follow standard design choices used in the literature (Rudin et al., 2021; Margolis et al., 2024). See Table 2 for details on the inputs received by our policy, where we consider the default observation noise levels in Isaac Lab (Mittal et al., 2023). Table 3 provides details on the reward function from Margolis et al. (2024) used for RL training. Note that $dt = 0.02$ seconds in our experiments, and we report total task returns normalized to $[0, 1]$ as our performance metric.

**RL training**    We apply PPO (Schulman et al., 2017) as the base RL algorithm for all of the methods considered in our experiments, and we follow the parallel training scheme proposed in Rudin et al. (2021). We collect data from 4,096 parallel training environments sampled from $c \sim \mathcal{C}_{\mathrm{ID}}$, and

Table 4: RL training details

| Hyperparameter | Value |
|---|---|
| **PPO hyperparameters** | |
| Parallel environments | 4,096 |
| Total updates | 10,000 |
| Update frequency (steps per environment) | 24 |
| Epochs per update | 5 |
| Minibatches per epoch | 4 |
| Discount rate ($\gamma$) | 0.99 |
| GAE parameter | 0.95 |
| Clipping parameter | 0.20 |
| Entropy coefficient | 0.01 |
| Maximum gradient norm | 1.00 |
| Initial learning rate | 1e$-$3 |
| Target KL | 0.01 |
| Optimizer | Adam |
| **Adversary hyperparameters** | |
| Policy updates between adversary updates | 10 |
| Adversary probability of intervention | 0.05 |
| Maximum adversary magnitude (m / s) | 1.00 |
| **Network architectures** | |
| Policy ($\pi$) MLP hidden layers | 512, 256, 128 |
| Adversary policy ($\tilde{\pi}_{\text{adv}}$) MLP hidden layers | 512, 256, 128 |
| Critic ($V^{\pi}$) MLP hidden layers | 512, 256, 128 |
| Context encoder ($f$) MLP hidden layers | 64, 64 |
| Context encoder latent feature size ($d$) | 8 |
| MLP activations | ELU |

apply RL updates after every $24$ steps per environment. We perform a total of $10,000$ updates during RL training, and we repeat this training process over 5 random seeds. RL training required approximately 2 hours of wall-clock time on a Linux workstation with Intel Core i9-13900K processors and a NVIDIA GeForce RTX 4090 GPU. As in Rudin et al. (2021), we apply an adaptive learning rate based on a target KL divergence, and we use Generalized Advantage Estimation (GAE) (Schulman et al., 2016) to train the critic. Observations are normalized using a running mean and standard deviation throughout the training process. We consider a Gaussian policy with a state-independent standard deviation. The policy mean, critic, and context encoder are all modeled using multilayer perceptrons (MLPs), and the policy standard deviation is parameterized separately. See Table 4 for a summary of the hyperparameters and network architectures used for RL training with PPO.

The current observation $s_t$ and latent feature vector $z_t = f(c_t)$ are provided as inputs to both the policy and critic in GRAM and contextual RL. For robust RL, only the current observation $s_t$ is provided to the policy and critic. For domain randomization, we consider two different implementations: a version that provides both $s_t$ and $z_t$ as inputs to the critic as done in contextual RL, and another that only provides $s_t$ as input as done in robust RL. In both cases, domain randomization considers the same policy input $s_t$ as robust RL.

For GRAM and robust RL, we also train an adversary policy $\tilde{\pi}_{\text{adv}}(\tilde{a}_t \mid s_t)$ with PPO. The adversary applies an external force to the robot's body in the xy plane, which causes an abrupt change to the robot's linear velocity. Adversary interventions occur randomly 5% of the time during data collection. The adversary policy $\tilde{\pi}$ is trained to output the direction of the force that minimizes performance ($\tilde{a}_t \in \mathbb{R}$ represents the angle that defines this direction in the xy plane), and the impact of the force on linear velocity is sampled from the range $[0, M_k]$ where $M_k$ is linearly increased from $0.0$ to $1.0$ meters per second over the course of training. We train the adversary by applying one adversary update for every 10 policy updates. The design choices used for adversarial training are motivated by Pinto et al. (2017) and Tessler et al. (2019).

Table 5: Adaptation module training details

| Hyperparameter | Value |
|---|---|
| **Training hyperparameters** | |
| Parallel environments | 4,096 |
| Total updates | 5,000 |
| Update frequency (steps per environment) | 24 |
| Epochs per update | 5 |
| Minibatches per epoch | 4 |
| Maximum gradient norm | 1.00 |
| Learning rate | 1e−3 |
| Optimizer | Adam |
| **Adaptation module hyperparameters** | |
| History length ($H$) | 16 |
| Epinet random input dimension ($m$) | 8 |
| Epinet random input samples per data point ($N$) | 8 |
| Minimum quantile for $\alpha$ finetuning ($u_{\min}$) | 0.90 |
| Maximum quantile for $\alpha$ finetuning ($u_{\max}$) | 0.99 |
| $\alpha$ value at maximum quantile | 0.01 |
| **Network architectures** | |
| Base network ($\phi_{\text{base}}$) MLP hidden layers | 512, 256, 128 |
| Epinet ($\phi_{\text{epi}}$) MLP hidden layers | 16, 16 |
| MLP activations | ELU |

**Adaptation module training** For GRAM and contextual RL, we apply a supervised learning phase after RL training where the adaptation module is trained for deployment using data collected with the student policy. We apply the same parallel data collection design used for RL training, and we perform a total of 5,000 updates to train the adaptation module. This training required approximately 1 hour of wall-clock time on a Linux workstation with Intel Core i9-13900K processors and a NVIDIA GeForce RTX 4090 GPU. We consider the most recent $H = 16$ steps of history as input to the adaptation module. As in Margolis et al. (2024), we model the adaptation module in contextual RL with an MLP, which takes as input a flattened version of the history $h_t$. We also apply this architecture for the base adaptation network $\phi_{\text{base}}$ in GRAM. Convolutional, recurrent, or transformer architectures are also possible, but we found that a simple MLP design was sufficient to achieve good performance in our experiments. See Table 5 for a summary of the hyperparameters and network architectures used for training the adaptation module.

For the robust adaptation module in GRAM, we consider an epistemic neural network using the epinet architecture from Osband et al. (2023). For $\phi_{\text{epi}}$ in (5), we follow the design choices proposed in Osband et al. (2023) and model the epinet as

$$\phi_{\text{epi}}(\tilde{h}_t, \xi) = (\eta_{\text{epi}}^L(\tilde{h}_t, \xi) - \eta_{\text{epi}}^P(\tilde{h}_t, \xi))'\xi, \tag{10}$$

where $\eta_{\text{epi}}^L(\tilde{h}_t, \xi), \eta_{\text{epi}}^P(\tilde{h}_t, \xi) \in \mathbb{R}^{m \times d}$, $\eta_{\text{epi}}^L$ is a learnable network, and $\eta_{\text{epi}}^P$ is a prior network with the same architecture as $\eta_{\text{epi}}^L$ but no trainable parameters. Intuitively, $\eta_{\text{epi}}^L$ learns to cancel out the effect of the prior network $\eta_{\text{epi}}^P$ for $\tilde{h}_t$ seen during training. We model $\eta_{\text{epi}}^L$ and $\eta_{\text{epi}}^P$ using small MLPs, and define $\tilde{h}_t$ to be the concatenation of $h_t$ and the output from the last hidden layer of the base network $\phi_{\text{base}}$. As in Osband et al. (2023), gradients are stopped in the calculation of $\tilde{h}_t$.

GRAM uses the epinet architecture for uncertainty estimation to distinguish ID training contexts from OOD scenarios at deployment time. Therefore, we are interested in quantifying uncertainty *relative* to the uncertainty estimates in ID contexts. We accomplish this by including a scale parameter $\beta$ and shift parameter $c$ in the calculation of $\alpha_t$ in (8), which we finetune at the end of training using a validation set of $\|\hat{\sigma}_\phi(h_t)\|^2$ values collected from ID contexts. We set the shift parameter $c$ to be the $u_{\min} = 0.90$ quantile of the validation set (note that $\alpha_t = 1.00$ when $\|\hat{\sigma}_\phi(h_t)\|^2 \leq c$), and we set $\beta$ such that $\alpha_t = 0.01$ at the $u_{\max} = 0.99$ quantile of the validation set. This allows $\alpha_t$ to identify when uncertainty is high at deployment time *relative* to ID contexts seen during training.

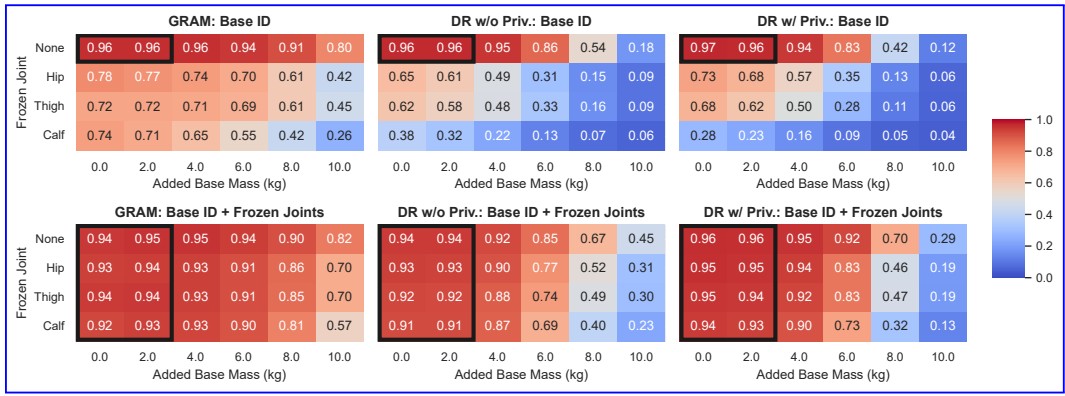

Figure 8: Detailed comparison of GRAM and domain randomization. Average normalized task returns across a range of added base masses and frozen joint types at deployment time. Black boxes represent ID training contexts contained in $\mathcal{C}_{\text{ID}}$. Top: Training with *Base ID* context set. Bottom: Training with *Base ID + Frozen Joints* context set. For both choices of $\mathcal{C}_{\text{ID}}$, domain randomization fails to generalize well outside of $\mathcal{C}_{\text{ID}}$ while GRAM achieves robust OOD performance.

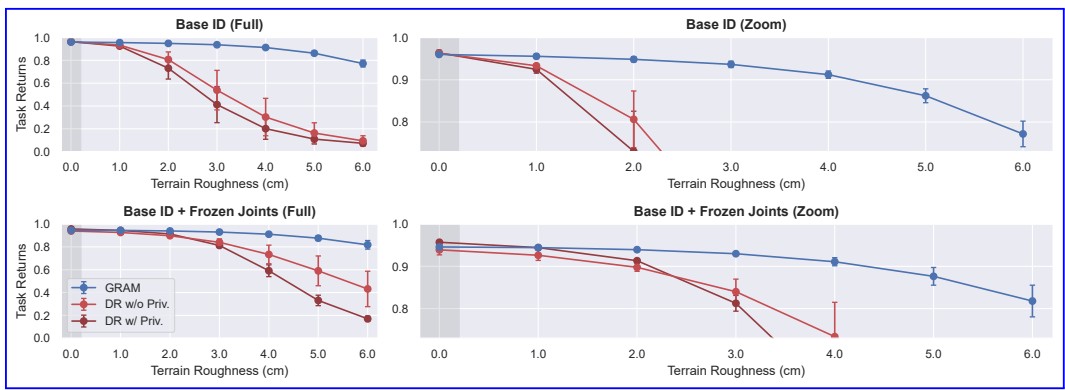

Figure 9: Detailed comparison of GRAM and domain randomization. Average normalized task returns across a range of terrain roughness at deployment time. Training only occurs on flat terrain (0 cm roughness), shaded in grey. Error bars denote standard deviation across 5 training seeds. Top: Training with *Base ID* context set. Bottom: Training with *Base ID + Frozen Joints* context set. Domain randomization does not generalize well to terrains not seen during training.

**Evaluation details** In our experiments, we evaluate the normalized task returns of the final trained policies across a range of deployment environments. Unless the parameter value is specified, performance is averaged across parameter values randomly sampled from the *Base ID* context set in Table 1. For each deployment environment considered in Section 7, performance is averaged over a total of 25,000 evaluation trajectories (5,000 evaluation trajectories per seed across 5 training seeds).

# B ADDITIONAL EXPERIMENTAL RESULTS

In this section, we provide detailed experimental results for the additional baselines shown in Figure 3, as well as experiments to analyze several implementation choices.

**Detailed domain randomization results** In Figure 8 and Figure 9, we provide a detailed comparison of GRAM to two implementations of domain randomization (with and without the use of privileged context information in the critic). We see that domain randomization is more robust in far-OOD scenarios when the critic does not incorporate privileged context information during training. Domain randomization provides slight OOD robustness benefits compared to contextual RL, but still

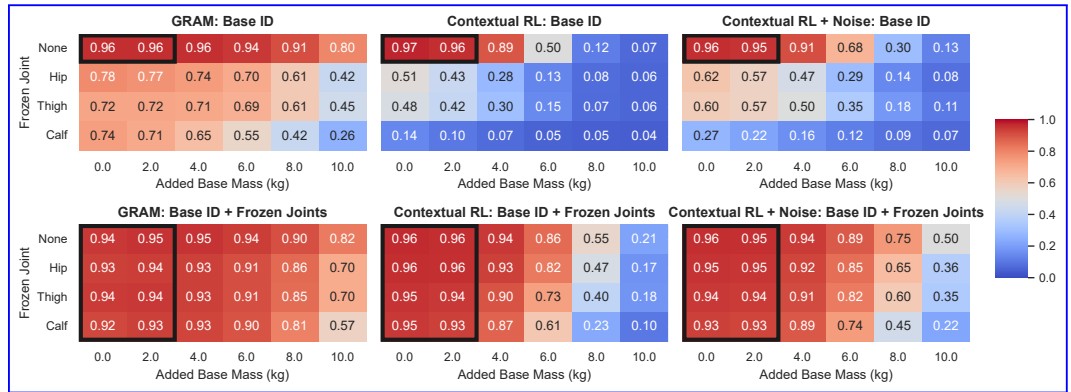

Figure 10: Detailed comparison of GRAM and contextual RL with random noise added to latent features during training. Average normalized task returns across a range of added base masses and frozen joint types at deployment time. Black boxes represent ID training contexts contained in $\mathcal{C}_{\text{ID}}$. Top: Training with *Base ID* context set. Bottom: Training with *Base ID + Frozen Joints* context set. For both choices of $\mathcal{C}_{\text{ID}}$, contextual RL with random noise is more robust than standard contextual RL in OOD scenarios but significantly less robust than GRAM.

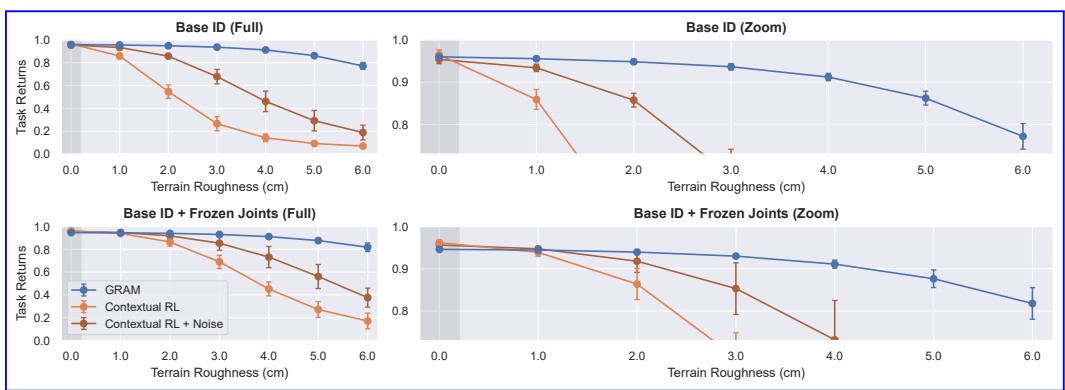

Figure 11: Detailed comparison of GRAM and contextual RL with random noise added to latent features during training. Average normalized task returns across a range of terrain roughness at deployment time. Training only occurs on flat terrain (0 cm roughness), shaded in grey. Error bars denote standard deviation across 5 training seeds. Top: Training with *Base ID* context set. Bottom: Training with *Base ID + Frozen Joints* context set. Contextual RL with random noise is more robust than standard contextual RL, but still fails to generalize well to far-OOD terrains.

fails to generalize well in OOD scenarios. In contrast, GRAM achieves robust OOD performance that is significantly better than domain randomization.

**Detailed contextual RL with noise results**   In Figure 10 and Figure 11, we provide a detailed comparison of GRAM to contextual RL with random noise added to latent features during training. In this baseline, we consider $z_t = f(c_t) + \sigma_z \epsilon$ during training, where $\epsilon \sim \mathcal{N}(\mathbf{0}_d, \mathbf{I}_d)$ and $\sigma_z = 0.25$. We set $\sigma_z = 0.25$ based on overall performance across a hyperparameter sweep of $\sigma_z = [0.25, 0.50, 1.00]$. The use of random noise is related to smoothing methods for robustness (Kumar et al., 2022), and we see in Figure 10 and Figure 11 that it does lead to improved OOD performance compared to contextual RL without latent feature noise. However, this approach is still significantly less robust than GRAM in OOD scenarios, further demonstrating the benefits of our framework.

**Analysis of history encoder training**   We follow the teacher-student training procedure proposed in Kumar et al. (2021) for training contextual RL and GRAM, where the history encoder $\phi$ is trained in a supervised learning phase using on-policy data collected by the student policy $\pi(a_t \mid s_t, \hat{z}_t)$

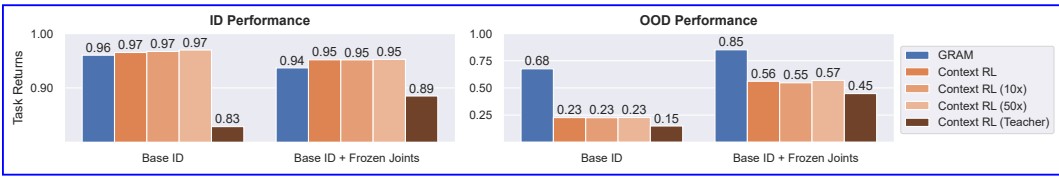

Figure 12: Impact of data source on history encoder training in contextual RL. Average normalized task returns in ID and OOD contexts. Performance averaged across contexts shown in Figure 4. Large replay buffers (10x and 50x the size of on-policy buffer) do not improve contextual RL performance. Training with data collected by the teacher policy instead of the student policy leads to a decline in contextual RL performance.

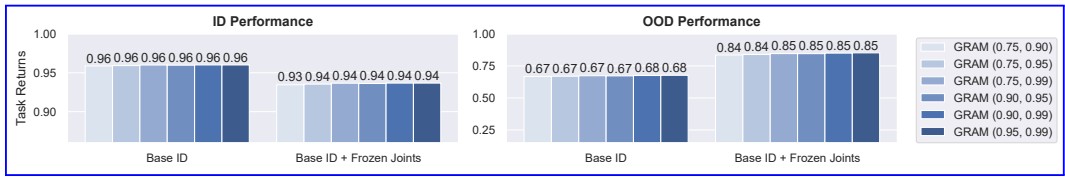

Figure 13: GRAM sensitivity analysis. Average normalized task returns of GRAM in ID and OOD contexts, using different validation quantiles $(u_{\min}, u_{\max})$ to finetune the scale parameter $\beta$ and shift parameter $c$ in the calculation of $\alpha_t$ in (8). Performance averaged across contexts shown in Figure 4. GRAM demonstrates consistent performance across different choices for finetuning $\alpha_t$.

with $\hat{z}_t = \phi(h_t)$. Using the parallel training scheme from Rudin et al. (2021), this results in a large and diverse batch of $\sim$100,000 samples per update. We analyze this implementation choice by considering contextual RL with larger replay buffers used for history encoder training, as well as contextual RL using the teacher policy for data collection during history encoder training.

For contextual RL with larger replay buffers, we reuse samples collected in the last 10 and 50 policy updates, respectively, resulting in replay buffers that contain $\sim$1 million and $\sim$5 million samples, respectively. For a fair comparison, we consider the same minibatch size and number of minibatch updates as the default setting by sampling from the replay buffer, so the only difference comes from how the data was collected. We see in Figure 12 that the use of a larger replay buffer does not improve the performance of contextual RL, which still performs significantly worse than GRAM in OOD scenarios.

We also compare against contextual RL using the teacher policy $\pi(a_t \mid s_t, z_t)$ with $z_t = f(c_t)$ for data collection during history encoder training. We see in Figure 12 that this choice leads to a performance decline for contextual RL, which may be caused by compounding errors in the history encoder that lead to a distribution shift between the data seen at deployment time with the student policy and the data collected during training with the teacher policy. This supports the use of a supervised learning phase with data collected by the student policy as proposed in Kumar et al. (2021).

**GRAM sensitivity analysis** For the GRAM coefficient $\alpha_t$, we finetune the scale parameter $\beta$ and shift parameter $c$ in (8) at the end of training based on the quantiles $(u_{\min}, u_{\max})$ of a validation set of $\|\hat{\sigma}_\phi(h_t)\|^2$ values collected from ID contexts. In our main experiments, we set $u_{\min} = 0.90$ and $u_{\max} = 0.99$. In Figure 13, we consider other choices of $(u_{\min}, u_{\max})$. We finetune the scale parameter $\beta$ and shift parameter $c$ in (8) with $u_{\min}, u_{\max} \in [0.75, 0.90, 0.95, 0.99]$ and $u_{\min} < u_{\max}$, and we find that the performance of GRAM remains consistent across these choices.

