# OpenReview forum: "GRAM: Generalization in Deep RL with a Robust Adaptation Module"
_ICLR.cc/2025/Conference — Submitted to ICLR 2025_

### Official Review · Reviewer_BE6V · 2024-10-30

**Soundness:** 3
**Presentation:** 3
**Contribution:** 2
**Rating:** 6
**Confidence:** 3

**Summary:**

This paper introduces GRAM (Generalization in Deep RL with a Robust Adaptation Module), a deep RL framework designed to enhance generalization performance in both in-distribution (ID) and out-of-distribution (OOD) scenarios. GRAM aims to unify both generalizations within a single architecture.

The core of GRAM is the robust adaptation module. This module utilizes an epistemic neural network to estimate the current context and the uncertainty associated with that estimation.  This uncertainty measure allows the system to identify OOD situations.  When high uncertainty is detected, the module defaults to a pre-defined "robust" latent feature, triggering a robust control policy specifically trained for such scenarios. This module is trained by teacher-student distillation. A "teacher" policy is trained with access to contextual information, learning to adapt to different environments.  A "student" policy then learns to mimic the teacher's behavior based only on observed history, enabling deployment without needing explicit contextual data.

GRAM's training pipeline also combines standard data collection for ID adaptation with adversarial training for OOD robustness.  An "adversary" agent introduces disturbances during training, forcing the robust policy to learn to handle unexpected perturbations.

Experiments conducted on simulated quadruped robot locomotion tasks demonstrate GRAM's effectiveness.  Compared to traditional contextual and robust RL methods, GRAM achieves stronger performance in both ID and OOD environments.  It demonstrates adaptive behavior in familiar terrains while maintaining robust locomotion in challenging, unseen terrains like rough surfaces.  The experiments also confirm that GRAM effectively identifies OOD situations, automatically adjusting its behavior based on the uncertainty level of its context estimation.

**Strengths:**

- GRAM tackles an important challenge of generalization in deep RL by proposing a unified architecture that effectively combines adaptation and robustness. This is an important perspective as previous methods often focused on one at the expense of the other.

- The introduction of the robust adaptation module with its integrated epistemic neural network is a novel mechanism. It allows the system to not only adapt to different contexts but also quantify the uncertainty of its estimations, enabling switching between ID and OOD modes.

- The paper is well-written and easy to follow.

**Weaknesses:**

### Major

- The assumption underlying this paper is notably strong. While assuming access to context variable values within the simulation environment is reasonable, *assuming the knowledge of which parameters constitute the context is a strong assumption*. In their experiments, contexts include friction, added base mass, motor strength, and joint angle bias - effectively acknowledging a priori which parameters will vary in test environments. This raises the question: if these variation sources are known beforehand, why not just randomly sample these parameters from a wider range during training?

- The adversary policy design similarly relies on prior knowledge. In their experiments, the adversary policy applies external forces to the robot's body. This is clearly tailored to address specific anticipated environmental variations. Such an adversary policy would be useless if the primary differences between training and test environments involved different factors, such as lighting conditions or camera poses.

- It seems the motivation of this paper is to deploy deep RL policies in real-world settings reliably (the 1st sentence in abstract). However, alll experiments are conducted in simulation.  While simulation allows for controlled experiments and extensive testing, demonstrating GRAM's effectiveness on real-world hardware is crucial for validating its practical applicability.  Real-world deployments often introduce unforeseen challenges that simulations may not fully capture.


### Minor

- The OOD experiments primarily focus on variations in terrain roughness and robot parameters. While relevant, exploring a wider range of OOD scenarios, such as unexpected obstacles or changes in task objectives, would provide a more comprehensive evaluation of GRAM's robustness.

- The paper mentions fine-tuning the parameters for the uncertainty hyperparameter ($\alpha_t$) but doesn't provide a detailed analysis of its sensitivity.  Understanding how different threshold values impact performance in various ID and OOD scenarios would be beneficial.

**Questions:**

- It is still unclear to me why the $z_{rob}$ is set to 0. Why should it output 0 for context embedding if uncertainty is high?

---

> ### Author Response · Authors · 2024-11-20
> **Author Response to Reviewer BE6V (1/3)**
>
> Thank you for your time and effort in reviewing our paper. It seems that there is a misunderstanding about the assumptions made in our approach:
>
> - **GRAM makes no assumptions about which parameters will vary at test time, and automatically adapts at test time to both in-distribution (ID) scenarios seen during training and out-of-distribution (OOD) scenarios not seen during training using only a history of observations and actions. Our experiments analyze both of these settings.**
> - **We utilize a common form of adversary policy for dynamics robustness that does not assume knowledge about the parameters that will vary at test time. We demonstrate the robustness of GRAM across a variety of OOD scenarios that are different from the structure of the adversary policy.**
>
> Please see below for detailed responses to your comments, which we hope clarify this misunderstanding and address your main concerns. We ask that you please consider updating your review scores to reflect these clarifications. Based on your suggestion, we have also updated the paper to include sensitivity analysis in Figure 13 related to the GRAM coefficient $\alpha$.
>
> ---
>
> > **The assumption underlying this paper is notably strong. While assuming access to context variable values within the simulation environment is reasonable, assuming the knowledge of which parameters constitute the context is a strong assumption. In their experiments, contexts include friction, added base mass, motor strength, and joint angle bias - effectively acknowledging a priori which parameters will vary in test environments.**
>
> It seems that there is a misunderstanding about the assumptions made by our algorithm. As mentioned above, **GRAM makes no assumptions about which parameters will vary at test time, and automatically adapts at test time to both ID scenarios seen during training and OOD scenarios not seen during training using only a history of observations and actions.**
>
> - We assume access to a set of ID contexts $\mathcal{C_\textnormal{ID}} \subset \mathcal{C}$ during training, and we refer to all other contexts as OOD contexts $\mathcal{C_\textnormal{OOD}}$. Our algorithm is specifically designed to achieve adaptive ID performance for $c \in \mathcal{C_\textnormal{ID}}$, and robust OOD performance for $c \in \mathcal{C_\textnormal{OOD}}$. Please refer to the Problem Statement in lines 115-128.
> - We follow the standard teacher-student training procedure used in Contextual RL, where the user defines the range of ID contexts $\mathcal{C_\textnormal{ID}}$ by selecting environment parameters to vary during training (e.g., the parameters in Table 1 for our Base ID setting). These parameters are provided as input to the context encoder $f$ during training (all other possible parameters that impact dynamics would be constant during training, so do not need to be provided as an input to the network). **This does not imply that these are the only parameters that may vary at test time.**
> - Our experiments include OOD scenarios that consider variations of parameters that are different from the ones varied during training. In Figure 5, we test on rough terrain despite only training on flat terrain. In the top row of Figure 4, we test on frozen joints even though the Base ID setting did not observe frozen joints of any kind during training.
>
> Your point highlights the main limitation of Contextual RL methods that we address in this work. Contextual RL often only works well within the range of ID contexts seen during training (as we show in our experiments), which are typically selected based on domain knowledge about likely test-time conditions. GRAM addresses this main limitation of Contextual RL by automatically identifying and reacting to OOD scenarios in a robust fashion at test time, while still retaining the ID adaptation capabilities of Contextual RL.

---

> > ### Author Response · Authors · 2024-11-20
> > **Author Response to Reviewer BE6V (2/3)**
> >
> > > **The adversary policy design similarly relies on prior knowledge. In their experiments, the adversary policy applies external forces to the robot's body. This is clearly tailored to address specific anticipated environmental variations. Such an adversary policy would be useless if the primary differences between training and test environments involved different factors, such as lighting conditions or camera poses.**
> >
> > - Our work focuses on dynamics generalization (we have updated the paper to emphasize this), and we consider a common choice of adversary used in the literature that provides external forces to the robot during training (e.g., [1, 2]). We perform tests across a variety of different types of OOD dynamics in our experiments to demonstrate the robustness of GRAM, all of which are different from the structure of the adversary policy (i.e., OOD scenarios do not involve applying external forces to the robot).
> > - Related to your comment, all robust / adversarial RL methods require some assumption on the type / level of robustness that is desired (i.e., either implicitly or explicitly defining an uncertainty set). We think it would be interesting to incorporate robustness across a range of different levels and types, which we have discussed in our Limitations and Future Work paragraph (see lines 529-532).
> > - Please note that lighting conditions and camera poses would not impact our experiments, which consider a policy based on proprioceptive observations available from standard onboard sensors (see Table 2). However, it would be interesting to extend our approach to address visual variations as you have mentioned. We have added commentary about this in our Limitations and Future Work paragraph (see lines 534-535).
> >
> > **References:**
> >
> > [1] Pinto et al. Robust adversarial reinforcement learning. In ICML 2017.
> >
> > [2] Xiao et al. PA-LOCO: Learning perturbation adaptive locomotion for quadruped robots. arXiv 2024.
> >
> > ---
> >
> > > **All experiments are conducted in simulation. While simulation allows for controlled experiments and extensive testing, demonstrating GRAM's effectiveness on real-world hardware is crucial for validating its practical applicability. Real-world deployments often introduce unforeseen challenges that simulations may not fully capture.**
> >
> > We have included a discussion about this choice in our Limitations and Future Work paragraph (see lines 535-538). As you have mentioned, we focused on simulated experiments in this work to conduct a comprehensive empirical study of GRAM across a variety of settings. We consider the realistic Isaac Lab simulator that includes sensor noise and has been shown to allow for zero-shot sim-to-real transfer (e.g., [2, 3, 4]). We agree that it would be very interesting to conduct real-world hardware experiments, and we are actively pursuing this as part of future work.
> >
> > Although we do not consider hardware experiments, we strongly believe that our work presents significant and novel contributions that are valuable to the community. We introduce a novel, unified algorithmic framework for ID and OOD dynamics generalization in deep RL, and provide comprehensive experimental evaluation on a complex, realistic simulation task across a variety of ID and OOD scenarios.
> >
> > **References:**
> >
> > [3] Kumar et al. RMA: Rapid motor adaptation for legged robots. In RSS 2021.
> >
> > [4] Margolis et al.. Rapid locomotion via reinforcement learning. The International Journal of Robotics Research, 2024.

---

> > > ### Author Response · Authors · 2024-11-20
> > > **Author Response to Reviewer BE6V (3/3)**
> > >
> > > > **The OOD experiments primarily focus on variations in terrain roughness and robot parameters. While relevant, exploring a wider range of OOD scenarios, such as unexpected obstacles or changes in task objectives, would provide a more comprehensive evaluation of GRAM's robustness.**
> > >
> > > - Our work focuses on dynamics generalization, and we believe that our experiments capture a broad range of OOD scenarios that impact dynamics in different ways (added base mass, frozen joints, rough terrain). We have updated the paper to further emphasize our focus on dynamics generalization.
> > > - We agree that applying our methodology to variations that require other modalities such as vision would be a very interesting direction, and we have added commentary about this in the Limitations and Future Work paragraph (see lines 534-535).
> > > - Multi-task RL represents an orthogonal line of research, where task information is typically known and can be provided to the policy as an additional input (similar to the velocity command input in our experiments). Our framework could be applied to provide dynamics generalization in a multi-task setting.
> > >
> > > ---
> > >
> > > > **The paper mentions fine-tuning the parameters for the uncertainty hyperparameter $\alpha_t$ but doesn't provide a detailed analysis of its sensitivity. Understanding how different threshold values impact performance in various ID and OOD scenarios would be beneficial.**
> > >
> > > Thank you for the suggestion. We set the shift and scale parameters in the calculation of $\alpha_t$ in (8) based on quantile values from a validation set of data collected from ID contexts. Please see lines 857-863 for details. Following your suggestion, we have also included experimental results for GRAM in Figure 13 in Appendix B where we set the shift and scale parameters using different choices of validation set quantiles. We see that the performance of GRAM remains consistent across these choices.
> > >
> > > ---
> > >
> > > > **It is still unclear to me why $z_{\textnormal{rob}}$  is set to 0. Why should it output 0 for context embedding if uncertainty is high?**
> > >
> > > The specific value of $z_{\textnormal{rob}}$ does not have physical meaning. Rather, $z_{\textnormal{rob}}$ defines a robust anchor point in latent space, and we train our policy to achieve robust performance when $z_{\textnormal{rob}}$ is provided as input (see the right side of Figure 2). Other values of $z_{\textnormal{rob}}$ are possible, but we believe that $z_{\textnormal{rob}} = \mathbf{0}$ is a simple and logical implementation choice for the reasons discussed in lines 242-248. This allows the context encoder $f$ to output values near $z_{\textnormal{rob}}$ at the start of training, and learn to move away from $z_{\textnormal{rob}}$ as needed to achieve adaptive ID performance. Our experiments also demonstrate that this implementation choice leads to strong ID and OOD performance.

---

> > > > ### Comment · Reviewer_BE6V · 2024-11-24
> > > >
> > > > Thank you for your detailed response!
> > > >
> > > > > GRAM makes no assumptions about which parameters will vary at test time
> > > >
> > > >
> > > > I understand this point, but my comment actually means the selection of relevant contexts for $\mathcal{C}$ indeed requires domain knowledge. For example, including friction rather than color in $\mathcal{C}$ stems from a prior understanding of physical dynamics. The real world presents infinite potential contexts, and identifying those potential factors relevant to generalization relies on prior knowledge about the test cases.
> > > >
> > > >
> > > > Overall, I feel although making such assumptions might be necessary for practical implementation, the paper's current presentation seems over-claiming.

---

> > > > > ### Author Response · Authors · 2024-11-24
> > > > > **Follow-Up Response to Reviewer BE6V**
> > > > >
> > > > > Thanks for your reply, which has helped us better understand your comment! Please see below for additional clarifications about this point, which we hope addresses your concern. We have re-read the paper and do not believe that we overclaim the scope of the work or our contributions. If there are specific lines where you think this is the case, please let us know and we would be happy to address them.
> > > > >
> > > > > **Scope of the paper:**
> > > > >
> > > > > - The focus of our work is **dynamics generalization**. By definition in our Problem Formulation (Section 3), context $c \in \mathcal{C}$ impacts the Contextual Markov Decision Process via the transition model $p_c = p(s’ \mid s, a, c)$ (see lines 106-107). We have also updated language throughout the paper to emphasize the focus on dynamics generalization.
> > > > > - We agree that visual variations such as color can be important in applications that consider vision inputs, but they are not the focus of this work as they do not impact dynamics. We have included commentary about this in the Limitations and Future Work paragraph (lines 534-535). Please also note that visual variations would not impact our experiments, which consider a policy based on proprioceptive observations available from standard onboard sensors (see Table 2).
> > > > >
> > > > > **Domain knowledge - ID vs. OOD performance:**
> > > > >
> > > > > - We agree that the choice of *in-distribution (ID)* contexts $\mathcal{C_{\textnormal{ID}}}$ to consider during training indeed involves domain knowledge. This allows the policy to be trained to achieve strong performance across what the user believes are likely test-time scenarios, and is a core component of most RL training procedures for real-world applications (i.e., what domain randomization parameters / values to consider during training).
> > > > > - We also agree that there are many possible unknown scenarios that can arise at deployment time in the real world, which is the motivation for our unified GRAM framework. GRAM is designed to achieve strong performance in user-defined ID scenarios $c \in \mathcal{C_{\textnormal{ID}}}$, and automatically deploys a more robust policy when $c \notin \mathcal{C_{\textnormal{ID}}}$ which is detected from history. The main limitation of Contextual RL is that it does not generalize well outside of the user-defined ID context set $\mathcal{C_{\textnormal{ID}}}$, and **this limitation is addressed by GRAM**.
> > > > > - We acknowledge that the OOD robustness of GRAM depends on the robust RL training procedure, which we have highlighted in our Limitations and Future Work paragraph (see lines 529-532). Our experiments demonstrate that GRAM achieves robust performance across a broad range of realistic OOD scenarios that impact dynamics in different ways and were not seen during training, while Contextual RL fails in these scenarios.
> > > > >
> > > > > We hope that these clarifications address your concerns. If so, we ask that you please consider updating your review scores based on these clarifications and the revised version of the paper. If you have additional questions, please let us know and we would be happy to provide further details before the end of the discussion period. Thanks again for your time and effort in reviewing our work!

---

> > > > > > ### Comment · Reviewer_BE6V · 2024-11-24
> > > > > >
> > > > > > Thank you for the response. I have updated the rating.

---

> > > > > > > ### Author Response · Authors · 2024-11-24
> > > > > > > **Thank you!**
> > > > > > >
> > > > > > > Thank you for your support of our work! We are glad that we have been able to address your concern.

---

### Official Review · Reviewer_GgR5 · 2024-11-03

**Soundness:** 3
**Presentation:** 3
**Contribution:** 3
**Rating:** 6
**Confidence:** 4

**Summary:**

This paper proposes a new method for robust in-context adaptation in reinforcement learning. To remedy the OOD context distribution in the inference, the GRAM method proposes to adapt the the latent vector with a pre-trained context encoder that estimate the uncertainty of the context and bias the latent context into a confident region (i.e., 0). Empirical results show that the adaptation yields robust performance w.r.t. other baselines in the frozen joint OOD test cases.

**Strengths:**

1. The method is simple, novel and easy to understand. It does not involve too much code change with vinilla in-context adaptation.
2. The empirical results is comprehensive in quadraped OOD generalization case, which is persuasive.

**Weaknesses:**

1. The paper does not consider other easy to implement baselines, e.g. domain invariance prediction, adding noise to the context, or use a larger replay buffer for the context encoder training. I think to show that this specific design is useful, one should also consider other easy-to-implement baselines.
2. GRAM biases towards 0, which is the mean at the beginning of the training. It is unclear if this is the best choice. How about bias toward the mean at the end of the training of the context encoder?

**Questions:**

See the two points in weakness.

---

> ### Author Response · Authors · 2024-11-20
> **Author Response to Reviewer GgR5**
>
> Thank you for your positive feedback on our paper. Based on your suggestions, we have incorporated additional experiments in the revised version of the paper (changes highlighted in blue). Please see below for detailed responses to your comments. If we have addressed your main concerns, we ask that you consider updating your review scores to reflect the revised version of the paper.
>
> ---
>
> > **The paper does not consider other easy to implement baselines, e.g. domain invariance prediction, adding noise to the context, or use a larger replay buffer for the context encoder training. I think to show that this specific design is useful, one should also consider other easy-to-implement baselines.**
>
> Thank you for these suggestions. We have added the following experiments to address your comment:
>
> - We include an additional baseline in In Figure 3 that adds random noise to the latent feature variable in Contextual RL, which is related to smoothing methods for robustness [1]. We provide detailed results for this baseline in Figures 10 and 11 in Appendix B, and discuss this baseline in lines 961-968. We see that adding random noise to the latent feature improves the robustness of Contextual RL, but GRAM remains significantly more robust than this baseline in OOD scenarios.
> - In Figure 12, we analyze the impact of using a larger replay buffer to train the history encoder $\phi$ in Contextual RL. Our main experiments follow the standard approach from [2], which uses on-policy data collected by the student policy. Using a parallel training scheme with 4096 environments, this results in a large and diverse on-policy batch of ~100k samples. We compare this to using larger replay buffers that store ~1M and ~5M samples, respectively (by reusing samples from the last 10 and 50 policy updates, respectively).  We see in Figure 12 that reusing data does not improve the performance of Contextual RL, which still performs significantly worse than GRAM in OOD scenarios.
>
> **References:**
>
> [1] Kumar et al. Policy smoothing for provably robust reinforcement learning. In ICLR 2022.
>
> [2] Kumar et al. RMA: Rapid motor adaptation for legged robots. In RSS 2021.
>
> ---
>
> > **GRAM biases towards 0, which is the mean at the beginning of the training. It is unclear if this is the best choice. How about bias toward the mean at the end of the training of the context encoder?**
>
> Other values of $z_{\textnormal{rob}}$ are possible, but we believe that $z_{\textnormal{rob}} = \mathbf{0}$ is a simple and logical implementation choice for the reasons discussed in lines 242-248. Our experiments demonstrate that this implementation choice leads to strong ID and OOD performance, so we did not experiment with other values. Note that $z_{\textnormal{rob}}$ is provided as an input to the policy for robust OOD training (see the right side of Figure 2), so it is not possible to consider a $z_{\textnormal{rob}}$ that depends on values of latent vectors at the end of training.

---

> > ### Comment · Reviewer_GgR5 · 2024-11-24
> > **Thanks for the Reply**
> >
> > Thanks for your reply. My concern on 1 is addressed.
> > I would highly suggest experiments with other $z_{rob}$.
> > Thus, I maintain my score.

---

> > > ### Author Response · Authors · 2024-11-24
> > > **Thank you!**
> > >
> > > We are glad that we have been able to address your concern #1 with additional baselines and experiments!
> > >
> > > For #2, we use the general notation $z_{\textnormal{rob}}$ in the paper to clearly present the intuition behind what the robust adaptation module accomplishes, and we propose the use of $z_{\textnormal{rob}} = \mathbf{0}$ to implement our GRAM framework. We have updated lines 244-245 to make this more clear. We have demonstrated that this leads to strong ID and OOD performance in our experiments, and do not believe that experiments with other $z_{\textnormal{rob}}$ would change the main contributions of the paper.
> > >
> > > Thank you again for your supportive feedback on our work!

---

### Official Review · Reviewer_zHHm · 2024-11-03

**Soundness:** 3
**Presentation:** 2
**Contribution:** 3
**Rating:** 6
**Confidence:** 4

**Summary:**

The paper develops an approach for generalization in reinforcement learning by tackling the problems of in-distribution generalization through contextual RL, and out-of-distribution generalization through robust RL. The authors develop a unified approach, called "Robust Adaptation Module" that relies of teacher-student training for tackling these challenges of generalization. The teacher policy is trained through RL in simulation by using privileged information of the environment that is not accessible at deployment. The student policy is trained to imitate rollouts from the teacher policy and leverage an adaptation module that predicts "context" from history of interactions. Experiments with a simulated quadruped robot in fixed velocity locomotion demonstrates the applicability of the approach in generalizing to different dynamics variations like mass and friction.

**Strengths:**

- The paper is interesting and the key ideas of adaptive RL and robust RL are easy to follow

- To the best of my knowledge the robust adaptation module relying on context identification from history and epistemic uncertainty of the policy is novel in the context of generalization in RL.

- The use of a teacher-student training paradigm is nice, and since the student doesn't require privileged information, the approach can be potentially scaled to real world tasks via sim2real.

- The experiments are on a non-trivial simulated quadruped locomotion task and the generalization to dynamcis variations corresponds to a practical use-case in controls and robotics.

**Weaknesses:**

- It is unclear how general is the proposed approach in dealing with different types of variations. The paper is motivated from the perspective of very generic generalization in RL but the experiments are limited to a single task/environment and correspond to only dynamics variations. What about variations in visual scenes? generalization to different tasks with the same robot?

- The in-distribution and out-of distribution generalization motivation is a bit confusing in the introduction. The authors should clarify scenarios where out-of-distribution generalization is possible without in-distribution generalization. The approach is based on the former not necessarily implying the latter and so this is important to motivate and explain with concrete examples.

- It is a bit confusing that context prediction module is only trained from on-policy data. Can't off-policy and generic offline data with the same robot/agent be used for learning this context prediction? This could potentially alleviate some of the challenges with out-of-distribution generalization as well and remove the need for the second part of the module.

- Some prior works that also do context prediction and demonstrate real-world quadruped locomotion through student-teaching training are not cited and discussed. For example the RMA paper below:

Kumar, Ashish, Zipeng Fu, Deepak Pathak, and Jitendra Malik. "Rma: Rapid motor adaptation for legged robots."

**Questions:**

Refer to the weaknesses above.

- It is unclear how general is the proposed approach in dealing with different types of variations. What about variations in visual scenes? generalization to different tasks with the same robot?

- The in-distribution and out-of distribution generalization motivation is a bit confusing in the introduction. What are examples of scenarios where out-of-distribution generalization is possible without in-distribution generalization?

- It is a bit confusing that context prediction module is only trained from on-policy data. Can't off-policy and generic offline data with the same robot/agent be used for learning this context prediction?

---

> ### Author Response · Authors · 2024-11-20
> **Author Response to Reviewer zHHm (1/2)**
>
> Thank you for the detailed comments and positive feedback. Based on your suggestions, we have updated the paper to emphasize our focus on dynamics generalization and include additional experimental results to support our implementation choices (changes highlighted in blue). Please see below for detailed responses to each of your comments. We ask that you please consider updating your review scores to reflect our responses and the revised version of the paper.
>
> ---
>
> > **[W1 / Q1] It is unclear how general is the proposed approach in dealing with different types of variations. What about variations in visual scenes? generalization to different tasks with the same robot?**
>
> - This paper focuses on the problem of dynamics generalization, where the context impacts the transition model (see lines 106-107). We have updated the language in the Abstract and Introduction to make this more clear at the beginning of the paper.
> - Please note that variations in visual scenes would not impact our experiments, which consider a policy based on proprioceptive observations available from standard onboard sensors (see Table 2). However, it would be interesting to extend our approach to address visual variations as you have mentioned. We have added commentary about this in our Limitations and Future Work paragraph (see lines 534-535).
> -  Multi-task RL represents an orthogonal line of research, where task information is typically known and can be provided to the policy as an additional input (similar to the velocity command input in our experiments). Our framework could be applied to provide dynamics generalization in a multi-task setting.
>
> ---
>
> > **[W2 / Q2] The in-distribution and out-of distribution generalization motivation is a bit confusing in the introduction. What are examples of scenarios where out-of-distribution generalization is possible without in-distribution generalization?**
>
> Robust RL is the main example of this scenario. Robust RL is designed to achieve robust out-of-distribution (OOD) performance, but does not perform in-distribution (ID) adaptation across different training environments. Instead, it outputs an action that maximizes worst-case performance, leading to overly conservative behavior in ID environments.
>
> ID generalization refers to the ability to identify and adapt to different environments seen during training to achieve the best performance. OOD generalization refers to the ability to achieve robust performance in environments that were not seen during training. See lines 33-38 in the Introduction, and lines 124-128 in the Problem Statement. Our algorithm GRAM achieves both adaptive ID performance and robust OOD performance.

---

> > ### Author Response · Authors · 2024-11-20
> > **Author Response to Reviewer zHHm (2/2)**
> >
> > > **[W3 / Q3] It is a bit confusing that context prediction module is only trained from on-policy data. Can't off-policy and generic offline data with the same robot/agent be used for learning this context prediction?**
> >
> > Our main experiments follow the standard approach from [1], which uses on-policy data collected by the student policy to train the history encoder $\phi$. Using a parallel training scheme with 4096 environments, this results in a large and diverse on-policy batch of ~100k samples. Based on your suggestion, we have added experimental analysis comparing the impact of using different types of data to train $\phi$ in Contextual RL.
> >
> > - In Figure 12, we analyze the impact of training the history encoder $\phi$ in Contextual RL using data collected with the teacher policy instead of the student policy, which would represent off-policy data. We see that Contextual RL performance declines when $\phi$ is trained using data from the teacher policy. This supports the use of a supervised learning phase for training $\phi$ using data collected by the student policy as proposed in [1].
> > - In Figure 12, we also analyze the impact of training $\phi$ in Contextual RL with off-policy data by considering a larger replay buffer of data from the student policy. We compare the on-policy approach to using larger replay buffers that store ~1M and ~5M samples, respectively (by reusing samples from the last 10 and 50 policy updates, respectively).  We see in Figure 12 that reusing data does not improve the performance of Contextual RL, which still performs significantly worse than GRAM in OOD scenarios.
> >
> > **References:**
> >
> > [1] Kumar et al. RMA: Rapid motor adaptation for legged robots. In RSS 2021.
> >
> > ---
> >
> > > **Some prior works that also do context prediction and demonstrate real-world quadruped locomotion through student-teaching training are not cited and discussed. For example the RMA paper [1].**
> >
> > We have indeed cited the RMA paper [1] when referencing the teacher-student architecture for ID adaptation in contextual RL (see lines 70 and 133, and the citation on lines 589-590). Our algorithm GRAM builds upon the teacher-student architecture used by RMA [1] and others (e.g., [2, 3]) to achieve both ID adaptation and OOD robustness within a single unified architecture.
> >
> > **References:**
> >
> > [2] Margolis et al.. Rapid locomotion via reinforcement learning. The International Journal of Robotics Research, 2024.
> >
> > [3] Lee et al. Learning quadrupedal locomotion over challenging terrain. Science Robotics, 2020.

---

> > > ### Comment · Reviewer_zHHm · 2024-11-24
> > > **Response to author comments**
> > >
> > > Dear authors,
> > >
> > > Thank you for answering the questions from reviewers. My question regarding relevant citations is addressed, but I am still not convinced by the generality of the proposed approach in dealing with different types of dynamics variations especially due to the lack of real-world results. As such I cannot strongly recommend accepting the paper - I will keep my weak accept score (since the paper is decent and I do not see any obvious flaws) and will not argue for acceptance in case any other reviewer argues otherwise.
> > >
> > > Thanks!

---

> > > > ### Author Response · Authors · 2024-11-24
> > > > **Thank you!**
> > > >
> > > > Thank you for your reply! We believe that our simulation experiments cover a broad range of realistic OOD scenarios not seen during training that impact dynamics in different ways (added base mass, frozen joints, rough terrain). GRAM achieves robust performance across these different scenarios, while the popular Contextual RL approach fails. We agree that additional validation of GRAM on hardware is a very interesting direction for future work. Nevertheless, we believe that our current work presents significant and novel contributions that are valuable to the community.
> > > >
> > > > Thank you again for the supportive feedback provided in your review and your weak accept review score!

---

### Official Review · Reviewer_E5e4 · 2024-11-04

**Soundness:** 3
**Presentation:** 3
**Contribution:** 3
**Rating:** 6
**Confidence:** 5

**Summary:**

The authors propose a novel method (GRAM) to learn RL policies that exhibit both adaptive and robust behavior through a unified training pipeline, specifically in the context of deployment environments that vary in dynamics at test time. GRAM employs context-conditioned policies where the context is automatically learned through a history of past observations and actions. Crucially, when the history falls OOD or is unable to provide context point estimates, the learned context converges to a special null context vector, so that robust behavior is triggered.

**Strengths:**

- The manuscript is extremely well written and provides clear statements to understand the proposed method.
- The authors demonstrate a successful implementation of recently introduced Epistemic Neural Networks for a relevant and important open problem in robot learning (i.e. generalization over unobservable environment contexts)

**Weaknesses:**

- Missing recent related works: the authors should consider citing and discussing recent works [1] and [2] as they present themselves as state-of-the-art methods in Robust RL and Domain Randomization as of 2024. Specifically, DORAEMON [2] tackles the same problem setting as in this work where privileged information is available at training time, and a history of previous observations and actions is used to allow implicit system identification at test time and promote adaptive behavior.

- Limited experimental evaluation:
  - While Domain Randomization is an extremely popular baseline for learning robust/adaptive behavior in sim2real settings, the authors provide little information to the implementation of this baseline and no comparison of GRAM vs. DR in fig. 4 and 5.
  - Implementation details of DR: I highly suggest the authors to compare GRAM against a DR baseline which also uses a history of previous state and actions, as this is the notorious way to implement DR [3]. Furthermore, DR may and should also leverage privileged information at training time, by using the notorious asymmetric actor-critic paradigm [2, 3, 4]. Conditioning the critic on the known context often drastically affects the results of DR methods vs. unprivileged critics, and does not require further assumptions.
  - The analysis is carried out on locomotion environments only, and simulated environments only. The generalization problem under unobservable dynamics is likely exacerbated and more challenging for manipulation and contact-rich settings, which would make the experimental evaluation more significant and relevant.

[1] Reddi, A. et al. "Robust Adversarial Reinforcement Learning via Bounded Rationality Curricula." ICLR 2024.

[2] Tiboni, G. et al. "Domain Randomization via Entropy Maximization." ICLR 2024.

[3] Peng, Xue Bin, et al. "Sim-to-real transfer of robotic control with dynamics randomization." 2018 IEEE international conference on robotics and automation (ICRA). IEEE, 2018.

[4] Handa, Ankur, et al. "Dextreme: Transfer of agile in-hand manipulation from simulation to reality." 2023 IEEE International Conference on Robotics and Automation (ICRA). IEEE, 2023.

**Questions:**

- Why is the performance of all baselines better in Fig. 3 (right) for the "Base ID + frozen joints" vs. the "Base ID" counterpart?

- Regarding the statement at the end of Sec. 5 "as the goal is to train the epinet in (5) to output estimates with low variance in ID contexts and high variance in OOD contexts.". Does GRAM promote the learned encoder to output high variance in OOD contexts at training time? Judging from Eq. (7) alone, it seems to me that the encoder is only trained to provide low variance on ID contexts, whereas a higher OOD variance is a spontaneous effect that occurs at test time only.

---

> ### Author Response · Authors · 2024-11-20
> **Author Response to Reviewer E5e4 (1/2)**
>
> Thank you for your supportive feedback and detailed suggestions that have helped to improve our paper. We have updated the paper to incorporate additional experimental results and implementation details based on your suggestions (changes highlighted in blue). Please see below for detailed responses to all of your comments and questions. We hope that we have addressed your main concerns, and we ask that you please consider updating your review scores to reflect the revised version of the paper.
>
> ---
>
> > **The authors should consider citing and discussing recent works [1] and [2].**
>
> Thank you for pointing out these interesting recent works. We have added references to these in the Related Work section. Note that the methods proposed in these works could be used within our GRAM framework: the robustness curriculum in [1] could be applied in our robust OOD training, and the domain randomization curriculum in [2] could be applied rather than considering a fixed set of ID contexts throughout training.
>
> **References:**
>
> [1] Reddi et al. Robust Adversarial Reinforcement Learning via Bounded Rationality Curricula. In ICLR 2024.
>
> [2] Tiboni et al. Domain Randomization via Entropy Maximization. In ICLR 2024.
>
> ---
>
> > **The authors provide little information to the implementation of domain randomization (DR) and no comparison of GRAM vs. DR in fig. 4 and 5.**
>
> In response to your comment, we have included additional implementation details related to our baselines in Appendix A (lines 796-801). We have also added detailed experimental results comparing GRAM and DR in Figures 8 and 9 in Appendix B, which contain the same analyses as Figures 4 and 5 in the main paper.
>
> ---
>
> > **I highly suggest the authors to compare GRAM against a DR baseline which also uses a history of previous state and actions, as this is the notorious way to implement DR. Furthermore, DR may and should also leverage privileged information at training time, by using the notorious asymmetric actor-critic paradigm.**
>
> Thank you for the suggestions.
>
> - Please note that Contextual RL represents a baseline that is trained across a variety of contexts, uses a history of previous states and actions as input to the policy (via the history encoder $\phi$), and uses privileged context information in the critic during training. This is one of our main baselines (which allows for ID adaptation), and GRAM extends this approach to also incorporate OOD robustness.
> - Our implementation of DR considers a policy that takes as input the current observation, which is a common baseline used in prior works that consider a teacher-student architecture (e.g., [3, 4]). We chose to compare against a version of DR that does not use privileged context information in the critic because we found that this choice led to slightly more robust performance. Based on your suggestion, we have also added as a baseline a version of DR that uses privileged context information in the critic. We have included this additional baseline in Figure 3, and provide detailed comparisons against GRAM in Figures 8 and 9 in Appendix B.
>
> **References:**
>
> [3] Kumar et al. RMA: Rapid motor adaptation for legged robots. In RSS 2021.
>
> [4] Margolis et al.. Rapid locomotion via reinforcement learning. The International Journal of Robotics Research, 2024.
>
> ---
>
> > **The analysis is carried out on locomotion environments only, and simulated environments only. The generalization problem under unobservable dynamics is likely exacerbated and more challenging for manipulation and contact-rich settings.**
>
> We chose to focus our experimental analysis on quadruped locomotion, which represents a complex task where Contextual RL has been applied for ID adaptation in prior works [3, 4]. We consider the realistic Isaac Lab simulator that includes sensor noise and has been shown to allow for zero-shot sim-to-real transfer [3, 4]. As mentioned in the Limitations and Future Work paragraph (lines 535-538), we focused on simulated experiments to conduct a comprehensive empirical study of GRAM across a variety of ID and OOD settings, and we plan to deploy our GRAM framework in real-world hardware experiments as part of future work.
>
> Note that our GRAM framework can be applied to a variety of applications where dynamics generalization is important, and we agree that it would be interesting to consider other tasks such as manipulation in future work. We have added commentary on this in the Limitations and Future Work paragraph (lines 538-539).

---

> > ### Author Response · Authors · 2024-11-20
> > **Author Response to Reviewer E5e4 (2/2)**
> >
> > > **[Q1] Why is the performance of all baselines better in Fig. 3 (right) for the "Base ID + frozen joints" vs. the "Base ID" counterpart?**
> >
> > The Base ID + Frozen Joints setting trains across a broader range of dynamics that includes scenarios where joints are frozen, while the Base ID setting does not observe frozen joint scenarios during training. As a result, the difference between ID and OOD dynamics is not as significant in the Base ID + Frozen Joints setting, which leads to better OOD performance for all algorithms. For a direct comparison of specific scenarios under the two different training distributions, please see Figures 4, 8, and 10 (compare top vs. bottom).
> >
> > ---
> >
> > > **[Q2] Does GRAM promote the learned encoder to output high variance in OOD contexts at training time? Judging from Eq. (7) alone, it seems to me that the encoder is only trained to provide low variance on ID contexts, whereas a higher OOD variance is a spontaneous effect that occurs at test time only.**
> >
> > You are correct. The robust adaptation module is trained to output low variance for ID contexts. Because it is trained on ID contexts but not on OOD contexts, OOD contexts will output higher variances than ID contexts. We have updated the statement to be more clear: “…as the goal is to train the epinet in (5) to output estimates with low variance in ID contexts.”

---

> > > ### Comment · Reviewer_E5e4 · 2024-11-24
> > >
> > > Thank you for the detailed response. All my questions have been thoroughly addressed.
> > > I may not raise my score because the work only presents experiments in simulation, which raises questions on how relevant GRAM can be for actual sim-to-real transfers and for which real-world tasks.

---

> > > > ### Author Response · Authors · 2024-11-24
> > > > **Thank you!**
> > > >
> > > > We are glad that we have been able to thoroughly address all of your questions! Hardware experiments are indeed an interesting avenue for future work, and the complex, realistic simulation experiments that we have conducted will allow us to directly test our learned GRAM policy on a Unitree Go2 quadruped robot in future work.
> > > >
> > > > Thank you again for your supportive feedback on our work!

---

### Author Response · Authors · 2024-11-20
**Author Response to Reviewers**

Thank you to all of the reviewers for their thoughtful feedback. We are encouraged by the positive comments that our paper is clear and easy to follow (all reviewers), proposes a novel framework to address an important problem (E5e4, zHHm, BE6V), and provides comprehensive experimental results (zHHm, GgR5).

We appreciate the highly constructive comments that we received, and we have updated the paper to incorporate reviewer suggestions. For your convenience, we summarize the main changes to the paper (changes highlighted in blue in the revised paper):

1. **Additional experimental results / baselines**: We have added several new experimental results in response to reviewer suggestions. This includes additional baselines (Figure 3), detailed results for all baselines (Figures 8-11), analysis supporting implementation choices (Figures 12-13), and additional discussion of implementation details (Appendix A lines 796-801, and Appendix B).
2. **Additional discussion and clarifications**: We have updated language to emphasize our focus on dynamics generalization, provided additional discussion on limitations and future work, and included additional references.

We have also replied directly to each reviewer with detailed responses. If we have addressed your main concerns, we ask that you please consider updating your review scores to reflect our responses and the revisions we have made to the paper. Thank you for helping us to improve our work!

---

### Meta-Review · Area_Chair_P6tY · 2024-12-29

**Metareview:**

The paper introduces GRAM, a generalization framework in deep reinforcement learning that uses a robust adaptation module to handle both in-distribution and out-of-distribution environment dynamics. The module identifies and reacts to different dynamics using a history of past observations and actions, and employs a joint training pipeline for both adaptation and robustness, which is demonstrated on simulated quadruped locomotion tasks.

The reviews were generally positive, noting that the paper was well-written, clear, and novel, and that it addressed an important problem in reinforcement learning. Reviewers found the robust adaptation module and the teacher-student training paradigm to be key strengths. However, some reviewers also raised concerns regarding the lack of real-world experiments, and the generality of the approach, which remained after the rebuttal.

In my own evaluation of the paper, the main weakness is the narrow evaluation -- either verification on the real robot, or expansion to another generalization domain would have been sufficient to make a convincing argument for the effectiveness of the method. Unfortunately, in its current state, the GRAM evaluation is prohibitively limited.

**Additional Comments On Reviewer Discussion:**

The authors responded to the reviewers' comments by adding experimental results, implementation details, and clarifying certain points in the paper. Overall, while the reviewers were generally satisfied with the responses and appreciated the authors' efforts to address their concerns, some reservations remained regarding the scope and practical applicability of the approach, particularly the lack of real-world experiments and the generality of the approach in dealing with different types of dynamics variations.

---

### Decision · Program_Chairs · 2025-01-22

Reject